# Single nucleus multi-omics regulatory landscape of the murine pituitary

Frederique Ruf-Zamojski [1✉], Zidong Zhang [2], Michel Zamojski[1], Gregory R. Smith[1], Natalia Mendelev [1], Hanqing Liu[3], German Nudelman[1], Mika Moriwaki [4], Hanna Pincas[1], Rosa Gomez Castanon [3], Venugopalan D. Nair[1], Nitish Seenarine[1], Mary Anne S. Amper[1], Xiang Zhou[5], Luisina Ongaro[5], Chirine Toufaily[5], Gauthier Schang[5], Joseph R. Nery [3], Anna Bartlett[3], Andrew Aldridge [3], Nimisha Jain[1], Gwen V. Childs [6], Olga G. Troyanskaya [2,7,8], Joseph R. Ecker [3,9], Judith L. Turgeon[10], Corrine K. Welt[4], Daniel J. Bernard [5] & Stuart C. Sealfon [1✉]

To provide a multi-omics resource and investigate transcriptional regulatory mechanisms, we profile the transcriptome, chromatin accessibility, and methylation status of over 70,000 single nuclei (sn) from adult mouse pituitaries. Paired snRNAseq and snATACseq datasets from individual animals highlight a continuum between developmental epigenetically-encoded cell types and transcriptionally-determined transient cell states. Co-accessibility analysis-based identification of a putative *Fshb* cis-regulatory domain that overlaps the fertility-linked rs11031006 human polymorphism, followed by experimental validation illustrate the use of this resource for hypothesis generation. We also identify transcriptional and chromatin accessibility programs distinguishing each major cell type. Regulons, which are co-regulated gene sets sharing binding sites for a common transcription factor driver, recapitulate cell type clustering. We identify both cell type-specific and sex-specific regulons that are highly correlated with promoter accessibility, but not with methylation state, supporting the centrality of chromatin accessibility in shaping cell-defining transcriptional programs. The sn multi-omics atlas is accessible at snpituitaryatlas.princeton.edu.

[1] Department of Neurology, Center for Advanced Research on Diagnostic Assays, Icahn School of Medicine at Mount Sinai (ISMMS), New York, NY, USA. [2] Lewis-Sigler Institute for Integrative Genomics, and Graduate Program in Quantitative and Computational Biology, Princeton University, Princeton, NJ, USA. [3] Genomic Analysis Laboratory, The Salk Institute for Biological Studies, La Jolla, CA, USA. [4] Division of Endocrinology and Metabolism, University of Utah, Salt Lake City, UT, USA. [5] Dept. of Pharmacology and Therapeutics, McGill University, Montreal, QC, Canada. [6] University of Arkansas for Medical Sciences, Little Rock, AR, USA. [7] Department of Computer Science, Princeton University, Princeton, NJ, USA. [8] Flatiron Institute, Simons Foundation, New York, NY, USA. [9] Howard Hughes Medical Institute, The Salk Institute for Biological Studies, La Jolla, CA, USA. [10] Department of Internal Medicine, University of California, Davis, CA, USA. ✉email: frederique.ruf-zamojski@mssm.edu; stuart.sealfon@mssm.edu

The pituitary gland plays critical roles in the modulation of key physiological functions, as it integrates regulation by the central nervous system with that of the endocrine system. It consists of anterior, intermediate, and posterior lobes. The anterior lobe that comprises ~80% of the gland, contains five major hormone-producing cell types (somatotropes, gonadotropes, lactotropes, thyrotropes, and corticotropes), as well as non-endocrine cells. Recent single-cell (sc) transcriptome studies highlighted a heterogeneity in pituitary cells and other molecular signatures that may reflect different functional cell states[1–5] in populations and sub-populations (also reviewed in[6] after the publication of our preprint[7]). Sc epigenetic assays, particularly methylome analysis, provide reliable identification of developmentally determined cell types, distinguishing them from potentially transient cell states that can be observed in transcriptome analyses[8].

Gene regulatory programs are generally orchestrated by transcription factors (TFs) via interaction with cis-regulatory genomic DNA sequences located in or around target genes. Epigenetic mechanisms, including changes in chromatin accessibility and DNA methylation (for review[9]), play crucial roles in the regulation of gene expression. Epigenomic profiling technologies have been developed to explore various layers of epigenetic regulation at sc resolution, including single nucleus (sn) ATACseq, which measures chromatin accessibility, and genome-wide mapping of DNA methylation[10,11]. The integration of sc epigenomics with sc transcriptomics provides the opportunity to elucidate the regulatory programs and epigenetic mechanisms underlying cell type-specific gene expression, and to resolve intercellular heterogeneity[12].

One limitation of sc sequencing technologies is that tissue dissociation can elicit artifactual gene expression[13–16]. Unlike sc-based methods, sn approaches are compatible with snap-frozen tissue samples and minimize ex vivo expression changes[17]. The transcriptome complexity identified by snRNAseq is comparable to that of scRNAseq methods[18]. Besides its reliability for profiling the transcriptome at sc resolution[13,15,19–21], sn isolation also allows mapping of the chromatin-regulatory landscape[22–24] and genome-wide measurement of DNA methylation[25].

In the present work, we sought to resolve transcriptional regulatory mechanisms in murine pituitary cells and characterize cellular heterogeneity at transcriptomic and epigenomic levels. We analyzed over 69,000 nuclei from individual snap-frozen male and female pituitaries for parallel analysis of both their transcriptome and genome-wide chromatin accessibility profiles. For sn DNA methylation profiling, we assayed over 2,000 nuclei from pooled snap-frozen male pituitaries. We show how the resulting sn multi-omics atlas provides new insight into regulatory network and gene control mechanisms that are relevant to pituitary physiology and disease.

## Results

### Sn multi-omics profiling in murine pituitaries.
In order to study the relationship between gene expression and epigenetic features, we generated a multi-omics sn atlas. Sn transcriptome and chromatin accessibility were assayed simultaneously in individual pituitaries using snap-frozen adult male and female murine tissue. Sn DNA methylation (snMethyl) was determined using pooled snap-frozen pituitaries from males (Fig. 1). From six individual pituitaries, a total of 35,707 nuclei were assayed by snRNAseq and 33,443 by snATACseq (Supplementary Fig. 1, Supplementary Table 1). 5-methylcytosine sequencing 2 data (snmC-seq2[25]) were obtained for 2,756 nuclei isolated from 30 male pituitaries (Supplementary Fig. 1, Supplementary Table 1).

Additional datasets were generated using alternate tissue handling, processing, and assay protocols (Supplementary Fig. 1, Supplementary Table 1, see "Methods"). These included different pituitary dissociation methods before cryopreservation, as well as sc vs. sn RNAseq assays. Snap-frozen pituitaries and gently dissociated cryopreserved pituitaries generated the highest quality sn or sc datasets for both RNAseq and ATACseq assays (Supplementary Fig. 1). To eliminate ex vivo changes, our analysis focused predominantly on data from the snap-frozen pituitaries. To reduce any batch effects, all snRNAseq and snATACseq libraries from same-sex snap-frozen pituitaries were generated simultaneously and all libraries were pooled for sequencing.

### Transcriptome analysis of the pituitary cell types.
We detected transcripts for an average of about 2,000 genes per nucleus in the ~6,000 nuclei sequenced per snap-frozen pituitary (Supplementary Table 2a). To improve the resolution of cell-type identification, all same-sex datasets were merged, and the cells were clustered using Seurat's shared nearest neighbor (SNN) method and visualized using t-Stochastic Neighbor Embedding representation (t-SNE; Fig. 2a, Supplementary Fig. 2a). The distribution of UMI counts (Supplementary Fig. 3a,d), mitochondrial gene expression (Supplementary Fig. 3b,e), and ribosomal gene content (Supplementary Fig. 3c,f) in merged male and in merged female datasets showed some variation in some clusters, such as stem cells, yet this did not influence the identification of cell type clusters. Batch effects were also unlikely to contribute to cell cluster composition because all libraries were pooled before sequencing, and the same-sex samples that were merged for analysis were processed simultaneously.

The cell clusters from snap-frozen pituitaries were annotated based on differential expression of key pituitary marker genes (see Fig. 2a for merged males and Supplementary Fig. 2a for merged females). In addition to the major pituitary cell types, we identified clusters comprising stem (progenitor) cells, proliferating cells, pituicytes, and various cell types associated with blood and vasculature, including immune cells (macrophages, leukocytes), endothelial cells, and pericytes. Interestingly, two clusters of stem cells were distinguishable (Fig. 2a and Supplementary Fig. 2a). These two clusters differed in their expression of several genes, including the neurotransmitter receptor genes *Gabrg3* and *Grin2a* (Supplementary Fig. 2c). We also identified two gonadotrope clusters, as well as a small cluster of *Pit1*-negative cells located between the corticotrope and the gonadotrope clusters, which most likely corresponded to undifferentiated corticotrope or gonadotrope cells (Fig. 2a and Supplementary Fig. 2a). We distinguished three distinct somatotrope clusters in males (Fig. 2a) and two in females (Supplementary Fig. 2a). Two clusters of lactotropes were observed only in females (Supplementary Fig. 2a).

### Chromatin accessibility analysis of the pituitary cell types.
We analyzed chromatin accessibility by snATACseq in the same male and female pituitaries assayed by snRNAseq (Fig. 1). An average of 5000 nuclei were analyzed per pituitary sample, yielding ~30,000 fragments per nucleus (Supplementary Table 2b). Same-sex datasets were merged, clustered, and visualized by t-SNE plots (Fig. 2b and Supplementary Fig. 2b). Clusters were identified based on open chromatin (i.e., peaks of accumulated reads) at the promoters of major cell type marker genes (Fig. 3). Eleven clusters were observed in males (Fig. 2b) and fifteen in females (Supplementary Fig. 2b). In females, the gonadotrope and lactotrope cell populations were distributed among two clusters, along with a proliferating lactotrope cluster. Analysis of chromatin accessibility data resolved subclusters of somatotropes both in males (Fig. 2b) and in females (Supplementary Fig. 2b).

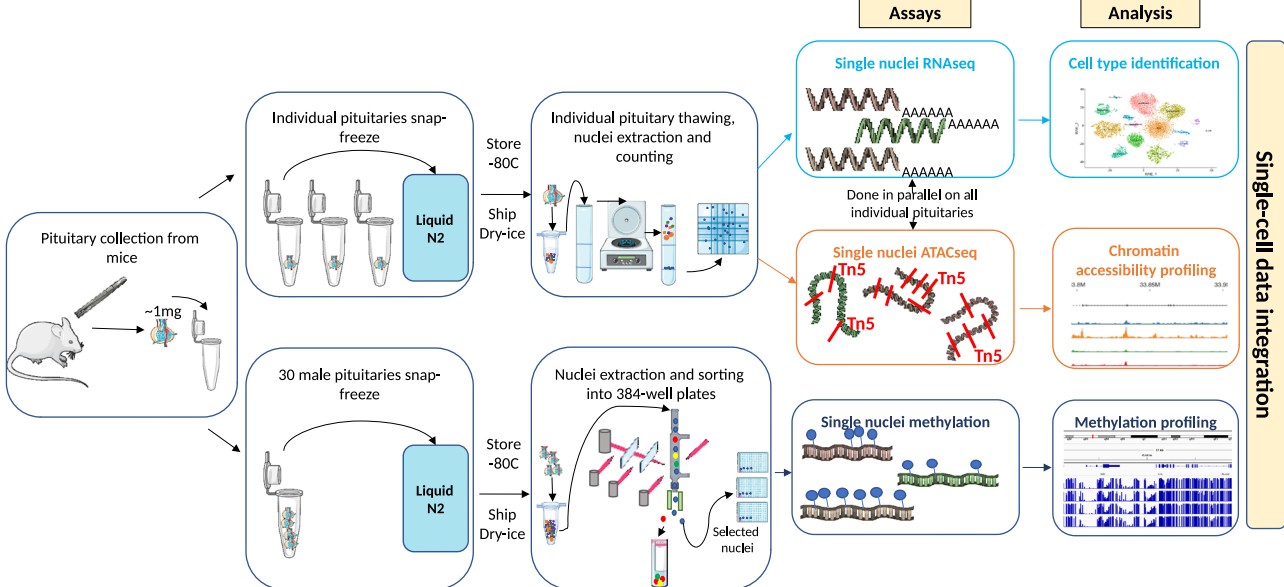

**Fig. 1 Overview of pituitary sn multi-omics experimental design.** SnRNAseq and snATACseq assays were performed in parallel on each individual snap-frozen murine pituitary following nuclei isolation from individual animals (top row). For snMethyl assays, pituitaries were pooled from 30 male mice, nuclei were isolated and sorted (bottom row). Data analysis was performed on each individual sample, as well as on merged same-sex samples to identify cell types before integrating all assay modalities and animals. Additional studies were performed on cryopreserved dissociated murine pituitary cells, including scRNAseq, snRNAseq, and snATACseq (see Supplementary Fig. 1 and Supplementary Table 1). Figure 1 was created using graphics from Servier Medical Art, licensed under a Creative Commons Attribution 3.0 Unported License https://creativecommons.org/licenses/by/3.0/.

The somatotrope clusters (Som) in males and females showed polarity in both snRNAseq and snATACseq, although the patterns leading to the gradients involved different genes. In the snRNAseq data, *Lhfpl3* and *Dpp10* were increased in one somatotrope cluster pole and *Tcerg1l* in the other (Supplementary Fig. 4a-d). However, the genes whose promoter accessibility differentiated the two poles in the snATACseq included *Rem1* for one pole, and *Ascl2* and *Oit1* for the other (Supplementary Fig. 5). The cluster labeled as Som0 in snRNAseq data from males and females was located between the lactotrope and the somatotrope clusters on the UMAP projection (Fig. 4, Supplementary Fig. 4). While the pattern of expression in these cells was distinct from the rare lactotrope/somatotrope doublets (Supplementary Fig. 4a), their expression of lactotrope, somatotrope, and lactosomatotrope genes, such as *Sgcz* and *Nrxn3*, suggests that this cluster may be comprised of lactosomatotropes (Supplementary Fig. 4a,e). These putative lactosomatotropes were not as distinctly separated in the snATACseq datasets. In females, we also observed a distinct cluster of lactotropes and somatotropes that showed chromatin accessibility for proliferating cell markers (Supplementary Fig. 2b) not seen in males (Fig. 2b). Accordingly, we labeled these two clusters as representing proliferating lactotropes and proliferating somatotropes (Supplementary Table 3). These data suggest that cell state and cell type form a continuum with gene expression and chromatin accessibility variances reflecting transient or more prolonged input from extracellular signaling.

**Methylation analysis of the pituitary cell types.** While the snRNAseq and snATACseq assays can be performed on single mouse pituitaries, the snMethyl method requires pooling pituitaries from many animals. Because of this limitation, we performed this assay only in males given the potentially increased complexity arising from estrous cycle stage differences among females. We obtained snMethyl data from 2,756 nuclei isolated from 30 pooled adult male pituitaries. Genome-wide levels of CG DNA methylation (mCG) were analyzed in 100 kb bins, clustered,

and visualized on t-SNE plots (Fig. 2c). Identified cell clusters were annotated to cell type based on mCG depletion at the promoters of known pituitary cell type markers. In contrast with brain[26,27], non-CG DNA methylation (mCH) was rare in the pituitary, and thus was not used for cell type assignment. The nine identified clusters comprised the major endocrine cell types, as well as stem/progenitor cells, endothelial cells, and pericytes (Fig. 2c). As expected, there was an inverse relationship between gene expression level (Fig. 2a, Supplementary Fig. 7) and DNA methylation level of the corresponding promoters (Figs. 2c and 3). Promoters with low methylation levels were more likely to be accessible and transcribed (see Fig. 3). In contrast with snRNAseq and snATACseq, the much smaller snMethyl dataset did not resolve any separate cell type subclusters.

**Comparison of sn and sc protocols.** Similar to snRNAseq analysis of snap-frozen pituitary samples, analysis of scRNAseq data from dissociated cells identified the major classical hormone-producing cell types (Supplementary Fig. 8). Clusters with low numbers of transcripts most likely corresponded to non-viable cells following cell dissociation (Supplementary Figs. 8, 9), as they were undetected in snap-frozen pituitaries analyzed by snRNAseq (Supplementary Figs. 10–12). Overall, the best quality RNAseq data based on sequencing metrics and cell-type identification were obtained from snap-frozen pituitaries (Supplementary Tables 1, 2).

Whether the ATACseq data were derived from nuclei from snap-frozen pituitaries (Supplementary Fig. 13) or cells from cryopreserved dissociated pituitaries (Supplementary Fig. 14), they exhibited a high signal-to-noise ratio at transcription start sites (TSS; see gene promoters in Supplementary Fig. 14). TSS enrichment scores for individual pituitaries ranged from 9.08 to 11.82, with an average of 10 for all 6 snap-frozen pituitary samples (Supplementary Table 2b). Data quality was comparable for all methods studied.

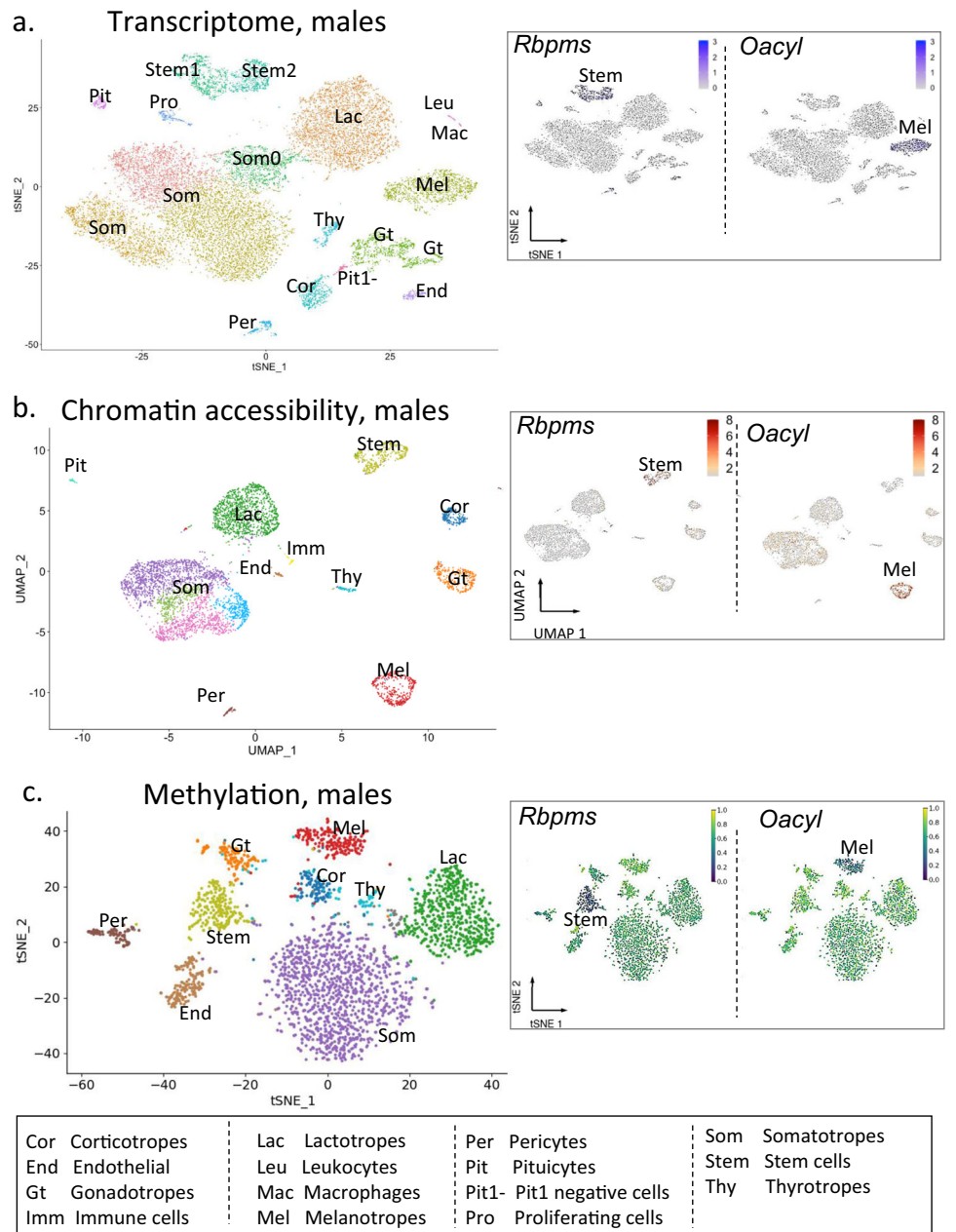

**Fig. 2 Identification of mouse pituitary cell types by merged sn analysis.** Each cluster is identified by a letter code. The code for each cell type is provided at the bottom of Fig. 2. **a** t-SNE representation of sn transcript expression in the merged datasets from three individual snap-frozen male pituitaries with cell clusters colored by type. Insert shows cell type identification based on established cell type markers for Stem cells (*Rbpms* expression) and for Melanotropes (*Oacyl* expression). **b** UMAP representation of sn chromatin accessibility in the merged datasets from the same three individual snap-frozen male pituitaries with cell clusters colored by type. Insert shows cell type identification based on established cell type markers for Stem cells (*Rbpms* chromatin accessibility) and for Melanotropes (*Oacyl* chromatin accessibility). **c** t-SNE representation of sn DNA mCG methylation in the dataset from a pool of thirty individual snap-frozen male pituitaries with cell clusters colored by type. Insert shows cell type identification based on established cell type markers for Stem cells (*Rbpms* promoter mCG methylation status) and for Melanotropes (*Oacyl* promoter mCG methylation status). Hypomethylation is associated with higher transcript expression (dark blue). Feature plots of the merged data for UMI counts, mitochondrial gene content, and ribosomal gene content are presented in Supplementary Fig. 3. Individual male and female sample t-SNE plots are presented in Supplementary Fig. 10, 13, and metrics are provided in Supplementary Table 2. Feature plots for UMI counts, mitochondrial gene content, and ribosomal gene content of the individual samples are presented in Supplementary Figs. 32–34. A scale is provided in each insert.

**Inter-animal and male–female variation in snRNAseq and snATACseq cell type clustering.** A snRNAseq library and a snATACseq library were prepared simultaneously from each individual pituitary. While general conclusions about cell type variation among these individual animals cannot be reached with datasets obtained from only three animals of each sex

(Supplementary Table 4), it was nonetheless important to explore whether the methods employed were able to define the transcriptome and accessibility pituitary landscapes of individual animals. The pituitary is a dynamic organ, especially in females, and obtaining detailed multi-omics data from single pituitaries sets the stage for future studies of transcriptional and epigenetic

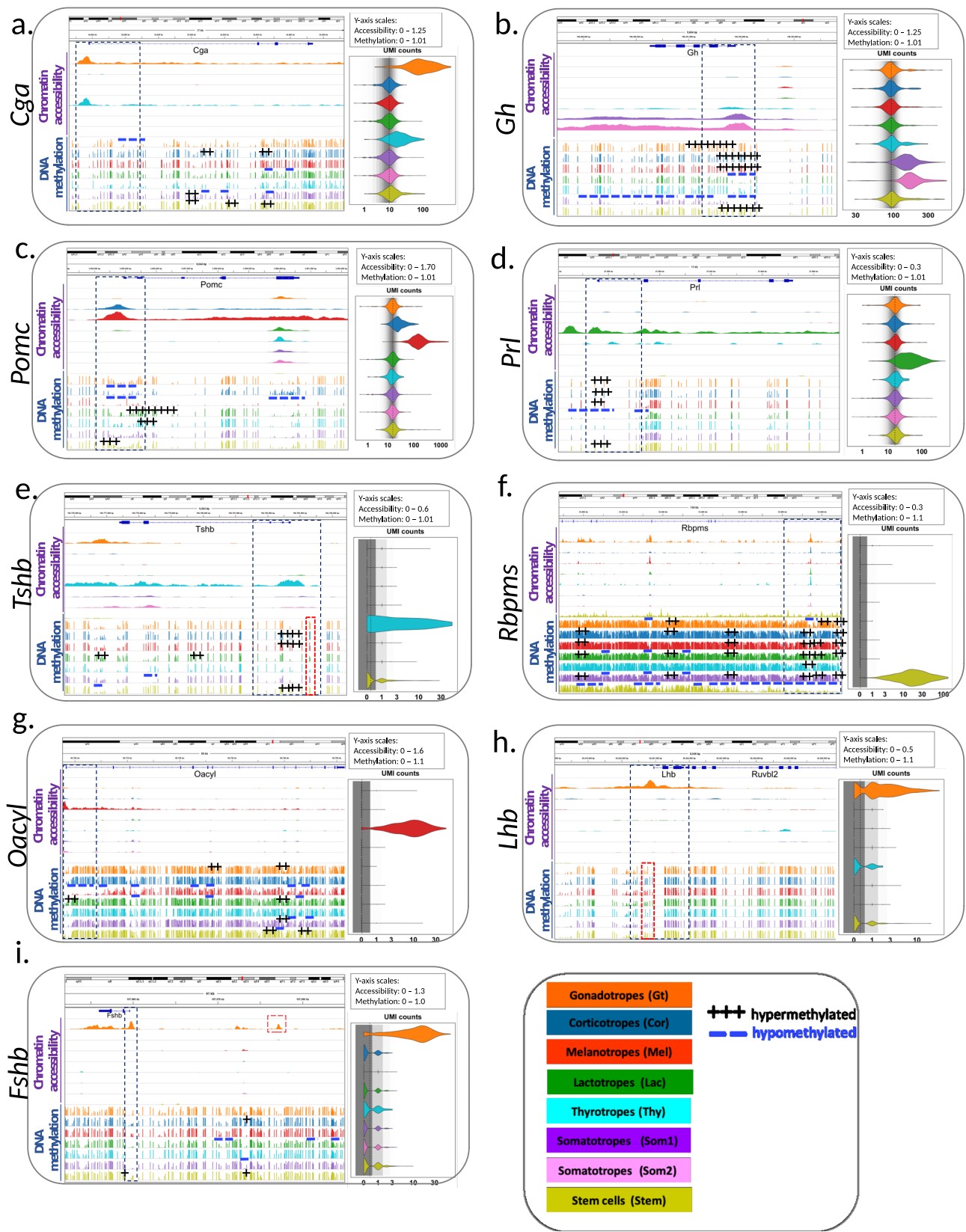

dynamics. We therefore investigated whether there was inter-individual variation in cell type clustering and whether such variation could be interpreted physiologically.

When the integrated snRNAseq data were labeled by individual animal, in males we could distinguish the somatotrope and the gonadotrope cluster originating from one animal from the somatotrope and gonadotrope clusters associated with the other two animals (Fig. 4a, compared with Fig. 2a). In contrast, labeling the snATACseq data obtained from the same pituitaries by animal showed no inter-individual differences in cell types

**Fig. 3 Multi-omics state of representative pituitary genes by cell type. a–i** Shown by cell type are genome browser tracks for chromatin accessibility (top), DNA methylation (bottom), and violin plots of transcript expression for the indicated genes (UMI count, right). Blue dotted boxes highlight the area around the promoter region. Red dotted boxes highlight a differentially methylated region (DMR; **e, h**) and/or open chromatin region (**i**) for the indicated genes. The two somatotrope subtypes of the main somatotrope cluster are presented, Som1 and Som2 representing the two poles (Refer to Supplementary Fig. 4). DMRs are indicated as either hypermethylated (+++), or hypomethylated (---) above each cell type track. Indicated at the bottom right corner is the order and color-coding for cell types and the methylation status.

(Fig. 4c). Although the sensitivity for animal-to-animal variation in the pooled male snMethyl dataset was limited, as pituitaries from 30 animals were combined and fewer than three thousand nuclei were assayed, no subclusters of the major cell types were observed, which might have indicated inter-animal differences (Fig. 2c). The additional gonadotrope and somatotrope clusters derived from the one male animal in the snRNAseq data were characterized by increased expression of immediate early genes (IEGs). Several other cell types from this animal that did not cluster separately, including lactotropes and melanotropes, also showed an increase in IEG expression (Supplementary Fig. 15). Previous studies of IEG induction following various perturbations in vitro and in vivo indicate that differential expression is not detected until at least 20 min[28–30]. In our study, the tissues assayed were snap-frozen within minutes of sacrifice. Therefore, the differences in IEG expression represent transcriptional changes that had occurred in vivo.

In females, the dataset from one of the animals contributed to an additional gonadotrope and lactotrope cluster observed in both snRNAseq and snATACseq datasets, as well as to an additional somatotrope cluster that was more clearly distinguished in the snATACseq data (Fig. 4b, d compared with Supplementary Fig. 2a,b). Both *Fshb* expression and chromatin accessibility at the *Fshb* gene body and promoter were increased in gonadotropes from this animal, as well as *Pitx1* expression, while key reproductive genes including *Greb1*, *Gnaq*, and *Pgr* were decreased (Fig. 4b, Supplementary Fig. 16). *Greb1* and *Crhbp* were decreased in lactotropes from this same animal as well (Supplementary Figs 16, 17). These results suggest that the multi-omics data have captured a different reproductive state in this animal, such as a non-cycling female or a female at a different estrous cycle stage. While some clusters arose only in a single female, individual animal labeling showed that the proliferative somatotrope and lactotrope clusters were shared by all three animals (Fig. 4d), thus representing separate somatotrope and lactotrope sub-types.

**Correspondence of cell type proportions across assay modalities.** Determination of cell type proportions in the snRNAseq data (Fig. 4e, Supplementary Table 2a) revealed that females had significantly more lactotropes and fewer somatotropes than males (Supplementary Tables 2-4). Similar cell type proportions were inferred from the snATACseq data obtained from the same pituitaries (Fig. 4f, Supplementary Tables 2b, 3b, 4b). The cell type proportions in males deduced from the snRNAseq and snATACseq datasets also corresponded to those obtained from the snMethyl analysis of a sample of pooled male pituitaries (Fig. 4g, Supplementary Table 2c). Generally, male-female differences in cell type proportions identified by either gene expression or promoter accessibility analysis were in accord, providing further support for the common identification of cell types across different omics modalities. When comparing all three pairs of assay modalities, the proportions of the major pituitary cell types showed high concordance ($R^2 > 0.98$; Fig. 4h).

**Integrated sn multi-omics analysis of the pituitary and resource.** We explored the multi-omics landscape of the main

pituitary cell types by integrating the data obtained by snRNAseq, snATACseq, and snMethyl. The detailed ATACseq signal and snMethyl tracks, and the expression levels of eight important pituitary markers are shown in Fig. 3.

In male gonadotropes, the promoters of *Cga* (Fig. 3a), *Lhb* (Fig. 3h), and *Fshb* (Fig. 3i) were accessible, hypomethylated, and all three mRNAs were expressed. Focusing on *Cga*, we found that its promoter was accessible both in gonadotropes and in thyrotropes (Fig. 3a). In addition, while DNA was hypomethylated in gonadotropes around the *Cga* promoter, it was hypermethylated in lactotropes, somatotropes, and stem cells (Fig. 3a). Due to the small number of thyrotropes, no region of their *Cga* methylation track was called as a differentially methylated region (DMR). Notably, DMR calling may be overly conservative when applied to a specific gene of interest, as it takes into account the false discovery rate arising from multiple hypothesis testing across all genes. Despite the limited number of thyrotrope cells, visual inspection of the *Cga* methylation track suggests hypomethylation in the same region of the gene that was significantly hypomethylated in gonadotropes. Consistent with the patterns of chromatin accessibility and DNA methylation, *Cga* mRNA levels were elevated in gonadotropes and thyrotropes (Fig. 3a).

In the *Lhb* gene, we detected a promoter region (red box, Fig. 3h), whose mCG levels appeared to be lower in gonadotropes relative to other cell types, although this region also was not flagged by the DMR algorithm. Interestingly, we discovered an open chromatin region upstream of the *Fshb* promoter only in gonadotropes (red box, Fig. 3i). Other pituitary cell types showed expected patterns of chromatin accessibility, methylation, and RNA expression for these key pituitary genes (Fig. 3).

Based on the assumption that higher chromatin accessibility and lower methylation levels result in higher gene expression, we overlaid the snRNAseq, snATACseq, and snMethyl data using the Seurat data integration pipeline, (Fig. 5a–c, Supplementary Table 6). Genome-wide chromatin accessibility and methylation data were converted to gene-level data and integrated with the transcriptomic data using nearest neighbor analysis. Good correspondence of the major cell type clusters was found for all assay modalities and datasets within each animal (Fig. 5a, b; Supplementary Table 6).

To provide an example of the use of the sn atlas to generate hypotheses for further study, we focused on the gonadotrope *Fshb* gene, which is essential for reproductive health. Patterns of co-accessibility between distal elements and their target promoter have been utilized to build genome-wide maps of cis-regulatory sequences from sc data[31]. We applied the Cicero co-accessibility framework to our snATACseq data to identify putative regulatory regions of individual genes. Co-accessible regions of classical pituitary gene markers were found within specific cell types (snpituitaryatlas.princeton.edu). In particular, we identified several co-accessible regions upstream of the *Fshb* gene promoter (Fig. 6a, Supplementary Fig. 18). One co-accessible region was gonadotrope-specific and was located 17 kb upstream of the promoter. The sequence of this putative regulatory region corresponded to the human single nucleotide polymorphism (SNP) rs11031006 (Fig. 6a, boxed region; see also Fig. 3i), which

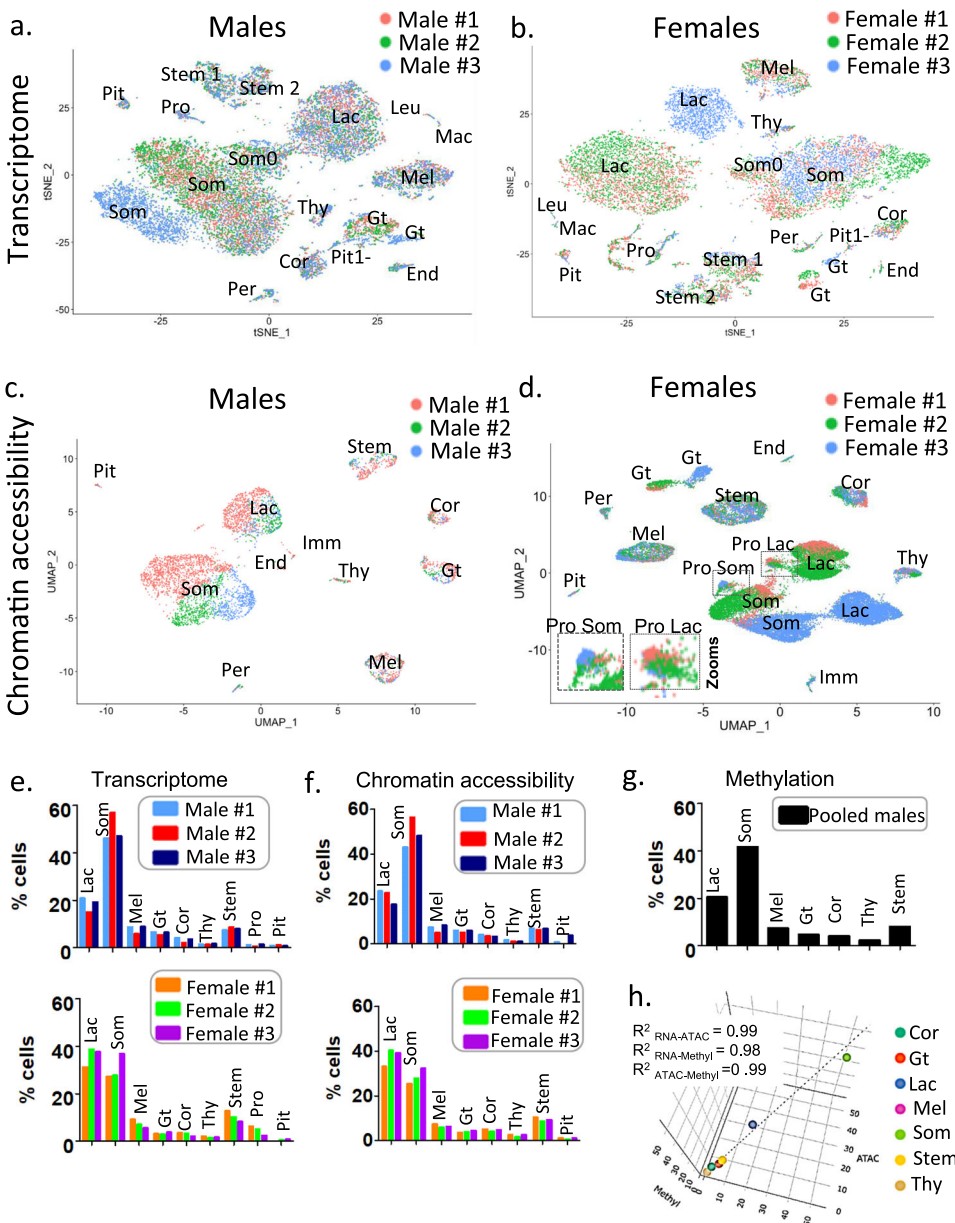

**Fig. 4 Male–female differences in cell types. a–d** t-SNE representation of sn transcript expression (**a**, males; **b**, females) and of sn chromatin accessibility (**c**, males; **d**, females) in merged same-sex samples, with labeling by individual animal. Individual animals are color-coded as indicated. Each cluster is identified by a letter code. The code for each cell type is presented at the bottom of Fig.2. In panel **d**, magnifications of the proliferative somatotrope and proliferative lactotrope clusters are shown as inserts. Refer to Fig.2a,b (males) and Supplementary Fig. 2a,b (females) for the merged analyses. **e, f** Cell type proportions per animal identified from the snRNAseq (**a, b**) and snATACseq (**c, d**) datasets show individual differences that are concordant (see also Supplementary Table 2). Male–female differences are significant for lactotropes and somatotropes ($p < 0.0001$) in both assays. $N = 6$ independent animals were examined (3 males, 3 females), with the following number of nuclei per assay and per animal: (i) for snRNAseq: male #1, 7,150; male #2, 4,562; male #3, 6,146; female #1, 6,717; female #2, 5,282; female #3, 4,186; (ii) for snATACseq: male #1, 2,782; male #2, 880; male #3, 1,095; female #1, 4,847; female #2, 9,976; female #3, 11,832. Data are presented per individual animal, with a single column for each. For the assessment of sex difference in cell type proportions, we used a two-way analysis of variance (ANOVA) followed by Bonferroni multiple comparisons post-hoc test, with $n = 3$ biological replicates per cell type. **g** Cell type proportions identified in snMethyl data from pooled male pituitaries. A pool of 30 animals was used with 2,756 nuclei analyzed. **h** 3D correlation plot between gene expression, chromatin accessibility, and snMethyl for the identification of cell type proportions in male samples.

has been linked to alterations in female fertility[32,33]. Multispecies comparative sequence analysis (PhyloP(60), PhastCons[34,35]) showed high sequence conservation of this domain across 60 vertebrate species (Fig. 6b, c). Together, co-accessibility analysis, sequence conservation, and GWAS studies identifying a fertility-linked human SNP at this location all support the assumption that this upstream domain contributes to the regulation of *Fshb*.

To test this hypothesis, we introduced CRISPR deletions in this region in the murine LβT2 gonadotrope cell line. *Fshb* expression and FSH secretion were undetectable in the parent gonadotrope cell line at baseline and following GnRH/activin A treatment. In contrast, deletion of the TTATTT sequence in this domain (Fig. 6c) led to an increase in baseline *Fshb* expression (Fig. 6d). Increased *Fshb* expression and FSH secretion were also observed

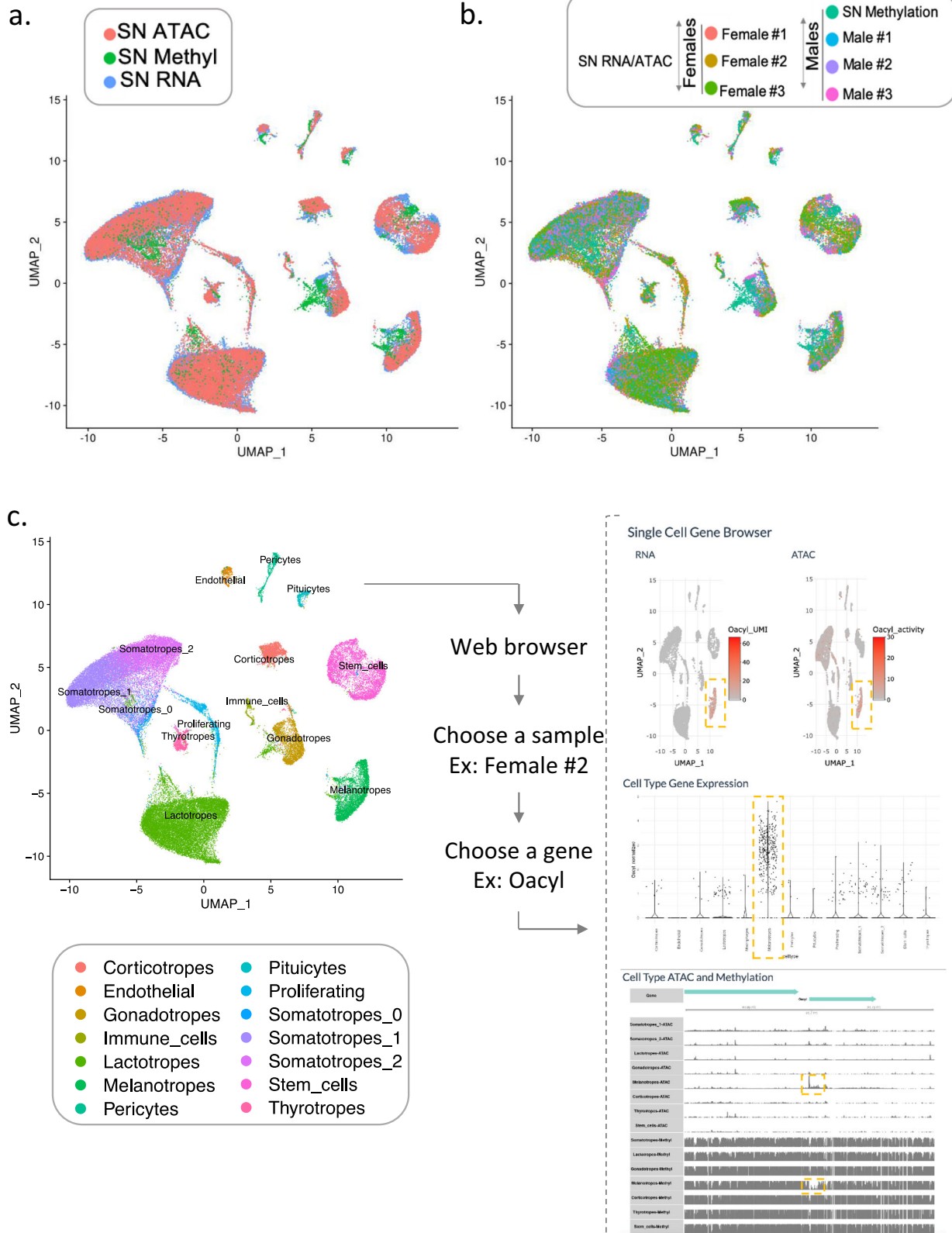

**Fig. 5 Multi-omics pituitary cell type integration. a** UMAP overlay of the snATACseq (red), snMethyl (green), and snRNAseq (blue) datasets. **b** UMAP of the overlay of all samples with the three omics modalities. SnRNAseq and snATACseq data are color-coded by animal. Male snMethyl data are also included. **c** Left panel, UMAP showing the cluster identification per cell type based on the snRNAseq annotations. Right panel, view of the web portal output. From the integrated datasets, users can access any information on gene expression, chromatin accessibility, and methylation status from our web portal snpituitaryatlas.princeton.edu.

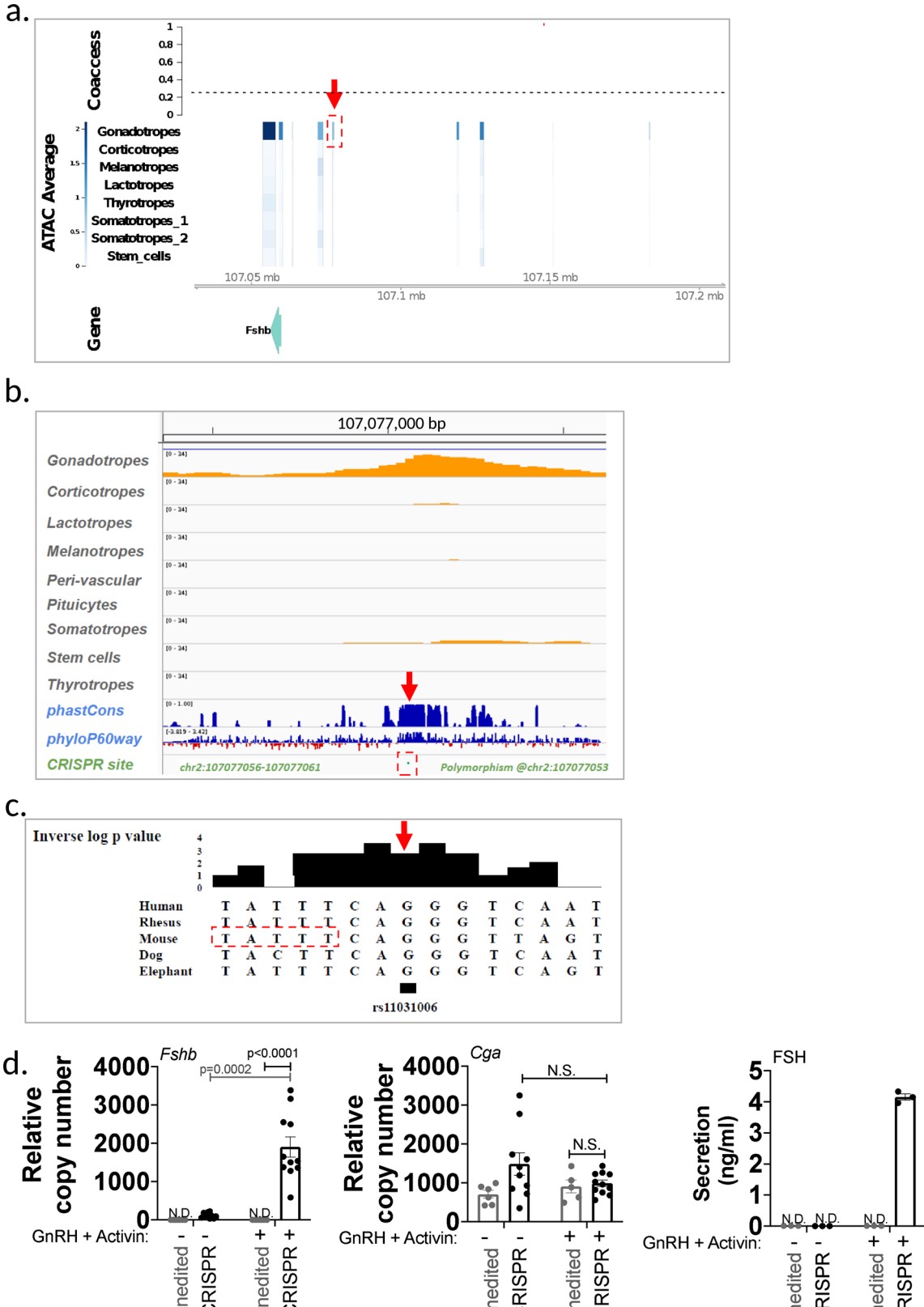

in this mutant following GnRH/activin A treatment. Two other CRISPR modifications at this site (i) a T deletion/TATT insertion; (ii) a TTT deletion) each led to a significant increase in *Fshb* expression following GnRH/activin A treatment, albeit to a lower extent than that observed in the TTATTT-deleted gonadotrope cell line (Supplementary Fig. 19). These findings suggest that the

reproductive phenotype associated with human SNP rs11031006 results from altered regulatory control of the *FSHB* gene via this co-accessible upstream domain, and support the value of co-accessibility analysis, available on the atlas web portal, for the identification of putative cis-regulatory domains for any genes of interest. After posting a preprint of our study[7], another group

**Fig. 6 Genome-wide inference of pituitary cis-regulatory domains:** *Fshb* **gene case study. a** Cis-regulatory domains of the *Fshb* gene inferred from snATACseq data correlation in a single male pituitary (analysis of all six individual pituitaries is presented in Supplementary Fig. 18). Boxed in red is the 17 kb upstream region corresponding to the human SNP rs11031006. **b** Conservation analysis of the boxed region indicated in (**a**). **c** Multi-species sequence alignment of the *FSHB* SNP rs11031006 (arrow). The boxed sequence was deleted by CRISPR. **d** Shown are *Fshb* (Left panel) and *Cga* (Center panel) gene expression, and FSH secretion levels (Right panel) in the unedited gonadotrope cell line (grey) and in the CRISPR deletion mutant line (black), either at baseline (-) or following GnRH and activin A treatment (+). All data points are shown. Biological replicates were $n = 4$ (*Fshb*), $n = 2$ (*Cga*), each measured in technical triplicates, and $n = 3$ (FSH). Among the 12 measurements for *Fshb*, 2 points were below sensitivity in the No treatment group and 1 in the treatment group. For *Cga*, one point was below sensitivity in the No treatment group. Shown are data from one representative experiment, out of 3 independent experiments with Treatment for both the edited and non-edited clone, 7 independent experiments for basal for the non-edited clones, and 5 independent experiments for basal of the edited clone. Data from additional CRISPR clones (unedited and edited) are illustrated in Supplementary Fig. 19. Error bars represent s.e.m. Significance was determined by two-way ANOVA with Bonferroni corrections. *N.S* not significant, $p > 0.05$; *$p < 0.1$; **$p < 0.01$; ****$p < 0.0001$.

independently validated our hypothesis, providing further support for the use of our atlas to identify clinically important new putative regulatory domains[36].

To help researchers utilize this sn mouse pituitary multi-omics atlas and to facilitate access to integrated data, we developed a web-based portal accessible at snpituitaryatlas.princeton.edu (Fig. 5c). Through this interface, users can access any gene from any pituitary cell type from each animal, and obtain the corresponding gene expression, chromatin accessibility, and promoter methylation information. The portal also provides easy access to inferred cis-regulatory regions and other integrative analyses.

**Coordinated sex- and cell-type-specific transcriptional and epigenetic programs.** We characterized coordinated gene expression and accessibility programs using our previously described Pathway Level Information ExtractoR framework (PLIER[37]). When applied to the snRNA expression or sn chromatin accessibility at gene promoters, PLIER identified sets of genes (Latent Variables, LVs) that varied in concert across different cell types or samples (Fig. 7a). PLIER incorporates information on known pathways and other gene set annotations to improve the identification of these LV sets and to associate them with biological processes. LVs are analogous to eigengenes[38] and can provide a summary level of overall expression or accessibility per cell type or sample, which can be displayed as a heatmap of the genes or promoters comprising the LV (Fig. 7a) or as a bar graph of the overall levels of each LV in each cell type (Supplementary Fig. 17).

PLIER analysis of the snRNAseq and snATACseq datasets identified LVs showing differential expression between sexes as well as LVs with predominant expression in each of the major pituitary cell types (Fig.7b, Supplementary Fig. 20). For example, LV1rna and LV1atac (Fig. 7b) showed increased expression/ accessibility in the female samples (Wilcoxon $p = 1.1e^{-5}$), specifically in female lactotropes (Wilcoxon $p = 0.02$). The pituitary from the same female that showed different gonadotrope and lactotrope clusters suggestive of a different reproductive state (see Fig. 4b, d), also showed divergent levels of LV1rna and LV1atac in lactotropes, with the LV1rna pattern appearing male-like. Correct sex identification of samples from all animals was confirmed by *Xist* and Y chromosome gene expression, as well as open chromatin accessibility on the Y chromosome for the males. We also compared LV1rna in scRNAseq data obtained from dissociated male and female pituitaries and confirmed the differential expression between females and males in this independent inquiry (Supplementary Fig. 17a, Wilcoxon $p = 1.1e^{-8}$).

Coordinated gene and chromatin accessibility programs were associated with all major pituitary cell types (Fig. 7b). Gonadotrope-predominant LVs are shown in Fig. 7b. Notably,

transcriptome-based LV2rna was elevated in gonadotropes ($p = 1.4e^{-5}$, Kruskal–Wallis test) and showed a similar expression level in all animals (Fig. 7b, left panel, Supplementary Fig. 21b, left panel). Gene accessibility-based LV2atac showed a similar pattern of increased gonadotrope accessibility ($p = 1.5e^{-6}$) among all animals (Fig. 7b, right panel, Supplementary Fig. 21b, middle panel).

We next examined the gene overlap between RNA and ATAC LVs that showed similar cell type patterns of enrichment (Fig. 7c). For example, the first 200 genes of LV1rna and the promoter accessibility sites comprising LV1atac significantly overlapped (hypergeometric test, $p = 3.6e^{-5}$). The significant overlap between the gonadotrope-enriched LV2rna and LV2atac genes (hypergeometric test; $p = 6.1e^{-15}$) also supports the formulation that they represent related transcriptional and epigenetic regulatory programs. In addition, when we applied the gonadotrope-enriched LV2rna directly to the snATACseq data, we also found a pattern of increased gonadotrope accessibility (Supplementary Fig. 21b, right panel; $p = 2.8e^{-5}$). Overall, PLIER analysis identified inter-sexual and inter-individual differences in related gene expression and promoter accessibility programs, as well as coordinated cell type-specific programs.

**Epigenetic control of cell type- and sex-specific regulons.** The physiology and biosynthetic responses of different cell types result from the activity of interacting gene regulatory networks (GRNs) or regulons, which are each targeted by a limited number of transcription factors (TFs). Single-cell datasets are valuable for the identification of the set of coregulated genes comprising each regulon and sharing binding sites for a TF driver[39]. As originally defined, regulons for each driver are identified across all cell types in a single-cell dataset. However, the composition of regulon genes expressed within each cell type, so-called cell-type-specific regulons, may differ (see Fig. 8a)[40]. We leveraged our multi-omics datasets both to characterize the pituitary regulons and to explore the role of epigenetic mechanisms in shaping the composition of cell type-specific regulons.

Analysis of our snRNAseq datasets using SCENIC[39] identified 344 pituitary regulons. The number of genes comprising each of these pituitary-wide regulons varied between 3 and 4,761, with a median size of 50. Determination of regulon activity in each cell type showed clusters of regulons with higher activity in different cell types (Supplementary Fig. 22a, Supplementary Table 7, snpituitaryatlas.princeton.edu). Many regulons were associated with TF drivers that were characteristic of pituitary cell types. For example, regulons showing high activity in the gonadotropes included TFs that are involved in gonadotrope development and/ or regulation of gonadotropin subunit gene expression (*Gata2*, *Foxl2*, *Nr5a1*, *Pitx1*, *Smad3*), or are gonadotrope-enriched (*Foxp2*; for review[1,41–43], Supplementary Fig. 22a). Because expression of a regulon in a cell depends on the expression of its driver, we

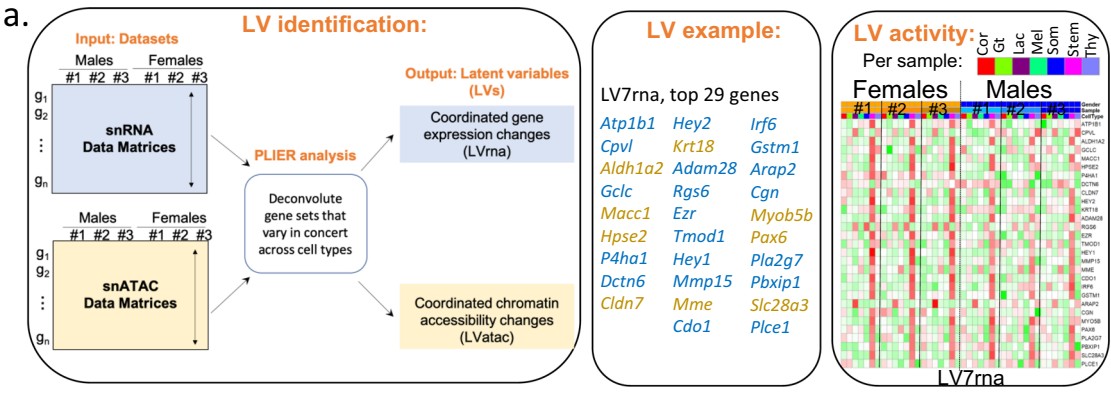

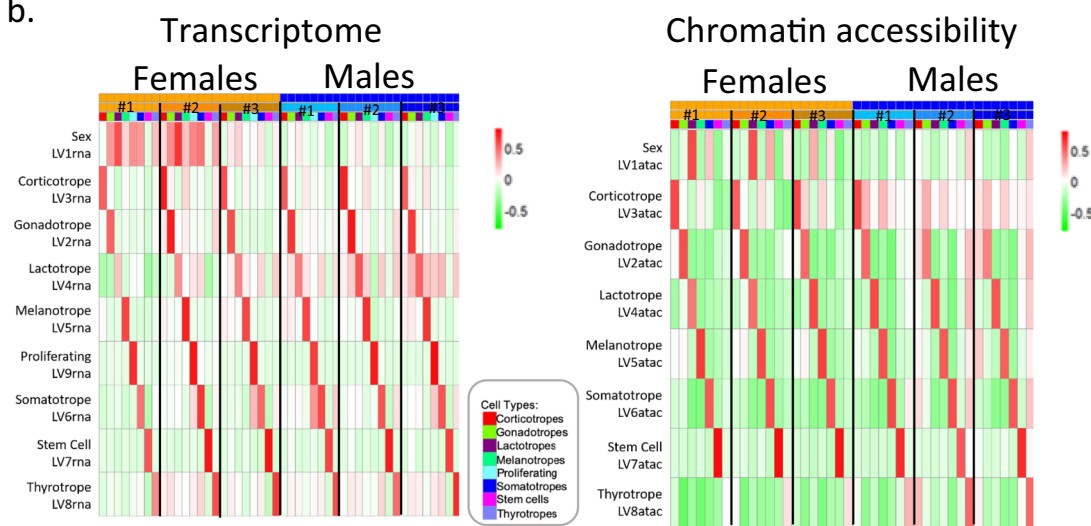

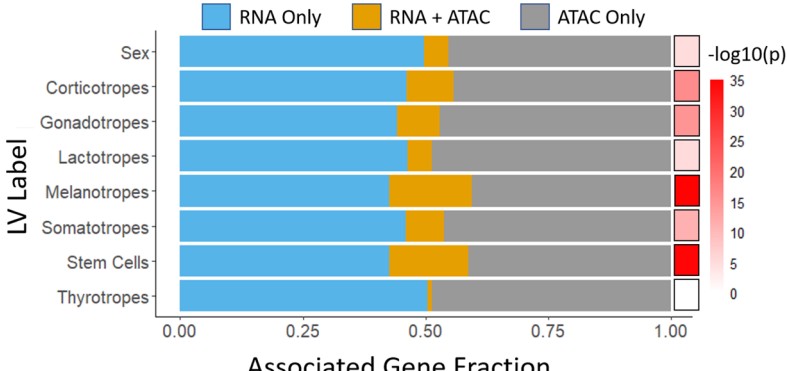

**Fig. 7 Cell type characterization based on specific coordinated gene expression and chromatin accessibility programs. a** Left panel, Schematic of PLIER functionality. SnRNAseq and snATACseq data matrices are generated for all six individual animals. In these data matrices, PLIER identifies sets of genes (latent variables, LVs) that show coordinated gene changes across different cell types or samples. These LVs are matched with publicly available information such as gene pathway sets. Middle panel, Example of the top 29 genes in LV7rna. Genes specific to LV7rna are in blue, while genes that are common between LV7rna and LV7atac are in gold. Right panel, Example of LV activity shows the expression levels of the top 29 genes in LV7rna for each pituitary cell type and individual animal. **b** Heatmaps of the levels of expression (Left) and chromatin accessibility (Right) of each LV for a specific animal/cell type, including LV7rna and LV7atac that are exemplified above. Each of the six individual samples is indicated. In the scale bar, red represents the highest level of gene expression or chromatin accessibility. **c** Graph showing the gene overlap (gold) between the RNAseq (blue) and ATACseq (grey) LVs for each cell type. The scale bar indicates the *p* values.

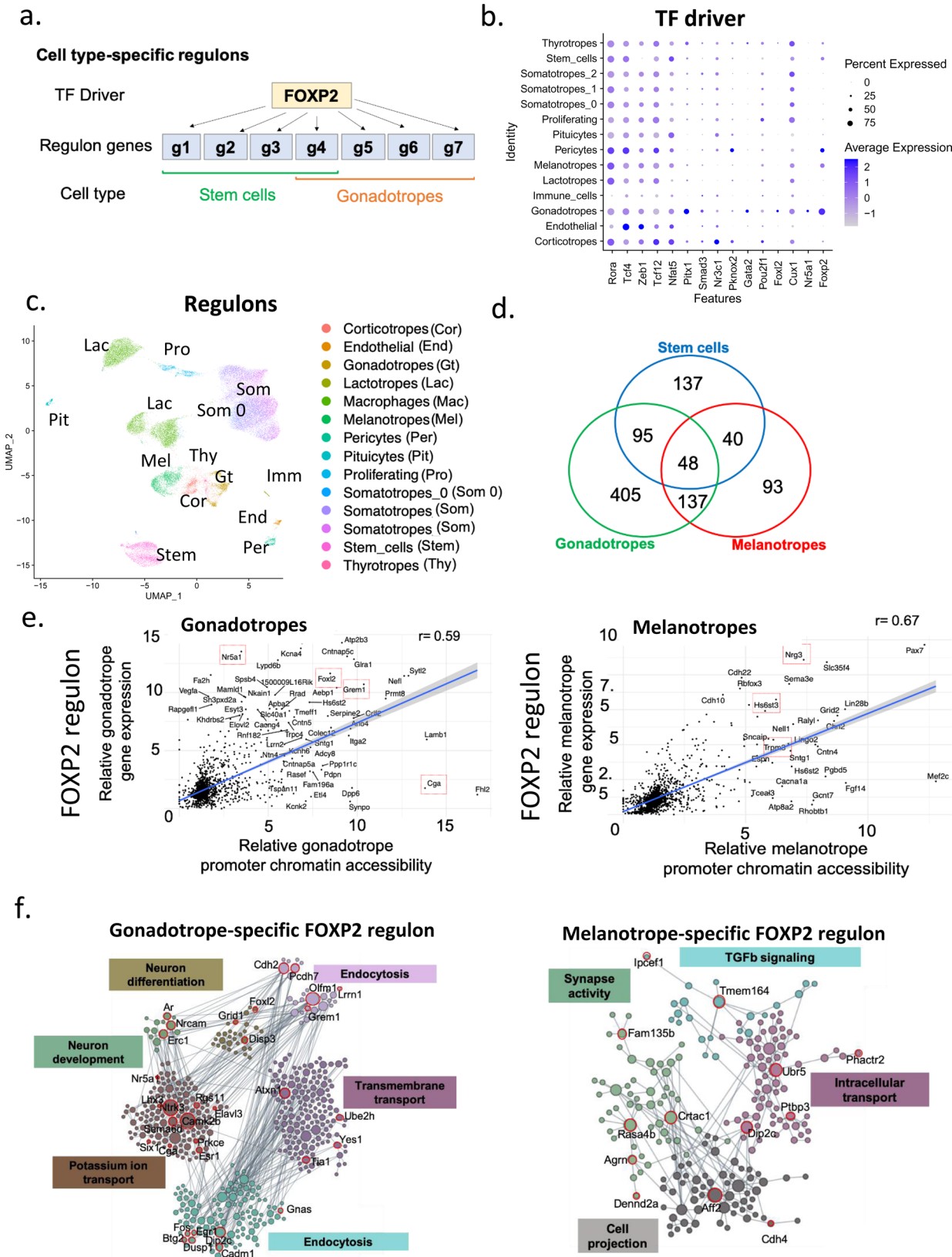

examined the differential expression of the driver genes in the different pituitary cell types. We found that some drivers are cell type-specific (*Foxl2* in gonadotropes), whereas others are expressed in several cell types (*Foxp2* in gonadotropes, stem cells, melanotropes, pituicytes, and thyrotropes; Fig. 8b).

We next studied whether the overall expression of the genes comprising each pituitary regulon was sufficient for cell-type classification. The overall level of activity of all the genes in each regulon within each individual cell was determined using the SCENIC pipeline and projected on UMAP axes (Fig. 8c,

**Fig. 8 Cell type-specific regulons. a** Schematic of a cell type-specific regulon, using FOXP2 as an example. A regulon consists of a TF driver (e.g., FOXP2) and its target genes. The regulon genes are cell type-specific (e.g., they differ in stem cells vs. gonadotropes). **b** Dot plot of the top regulons in male gonadotropes and their relative expression in other cell types. The color scale is indicated. The dot size symbolizes the percentage of cells expressing that regulon. **c** UMAP and cell type identification from the SCENIC regulon analysis recapitulates the clusters identified by data integration (Fig.5c). **d** Venn diagram of the overlap of FOXP2 regulon genes between the different cell types in males. **e** Scatter plots showing the correlation between relative promoter accessibility and relative gene expression for the 1,676 genes composing the FOXP2 regulon in male gonadotropes (Left) and male melanotropes (Right). Characteristic gonadotrope and melanotrope genes are framed in red. **f** Annotated gene modules showing the different functions associated with the FOXP2 regulon in gonadotropes (Left) and in melanotropes (Right). Refer to Supplementary Fig. 24 for the corresponding analysis of female samples.

Supplementary Fig. 22a). This projection, which is based solely on regulon gene set activity levels clustered all the major cell types identified by Seurat analysis of the snRNAseq data (compare with Fig. 5c). Regulon-based clustering separated lactotropes into three clusters. Annotation of these clusters by individual animals showed that the clustering was based on differences in lactotrope regulon activity between males and females as well as between individual females (Supplementary Fig. 22b). The male–female and inter-individual regulon-based cell cluster differences corresponded with similar differences observed after individual sample analysis of RNA expression levels (Fig. 4).

We examined whether epigenetic mechanisms could contribute to the gene composition of cell type-specific regulon expression. We first considered the FOXP2 regulon, which was expressed in several pituitary cell types (Fig. 8b). The gene composition of the FOXP2 regulon was dramatically different in gonadotropes, melanotropes, and stem cells (Fig. 8d). To study the role of epigenetic mechanisms in shaping cell type-specific regulon expression, we analyzed the correlation between promoter methylation or promoter chromatin accessibility and RNA expression of each of the 1676 genes comprising the pituitary FOXP2 regulon. Promoter methylation was poorly correlated with cell type-specific gene expression (Supplementary Fig. 23 c-e). In contrast, promoter accessibility of all the FOXP2 regulon genes in each of the three cell types was highly correlated with RNA expression (Fig. 8e and Supplementary Fig. 23a, 24). We next explored the biological processes regulated by the FOXP2 cell type-specific regulons using functional network module analysis ([44]HumanBase https://hb.flatironinstitute.org). Notably, the genes comprising each of the three cell-type-specific FOXP2 regulons showed modules with different functional annotations (Fig. 8f, Supplementary Fig. 23b). Thus, the three cell-type-specific FOXP2 regulons were shaped by differences in the chromatin accessibility landscape and modulated distinct biological processes in each cell type.

To our knowledge, sex differences in regulons within the same cell type have not been previously reported. We identified such sex-specific regulons and studied their epigenetic characteristics. The FOXL2 regulon, expressed only in gonadotropes (Fig. 8b), differed between males and females. As was the case for cell-type-specific regulons, we found that the relative levels of gene expression and promoter chromatin accessibility in the gonadotropes for all 331 FOXL2 regulon target genes were highly correlated in males (Supplementary Fig. 22c) and in females (Supplementary Fig. 25a). Generally, genes without open promoters were not transcribed, whereas genes with open promoters were transcribed, with varying levels of gene expression (Supplementary Fig. 22c). To determine whether this differential chromatin accessibility correlated with DNA methylation changes, we examined the promoter methylation levels of the regulon genes in males. While relative methylation levels of the regulon genes showing high vs. low relative expression in gonadotropes did not vary significantly ($p = 0.6$), comparison of the highest and lowest expressed genes showed modest statistical significance ($p = 0.008$; Supplementary Fig. 26). Functional

module analysis showed these sex-specific regulons were annotated to different processes in males and females (Supplementary Figs. 22e,f, 25b).

Like *Foxl2*, *Nr5a1* expression was largely restricted to gonadotropes (Fig. 8b), and the genes comprising this regulon differed in gonadotropes from males and females. Promoter accessibility was also a major determinant of the composition and functional pathways of the NR5A1 sex-specific regulons in these cells (Supplementary Fig. 27). The correlation between regulon gene expression and promoter and gene body methylation levels was not significant (Supplementary Fig. 27e). These results indicate that sex-specific regulons encompass different biological functions in males vs. females, and that promoter accessibility is a fundamental determinant of the sex differences in regulon activity. Based on these results, we propose a model for the cell type-specific regulation of regulon composition and function (Fig. 9).

## Discussion

We generated an integrated, sn multi-omics resource to elucidate the epigenetic mechanisms that regulate transcriptional networks in the pituitary. Our study identified epigenetically-defined cell type composition, cell type-specific and sex-specific differences in transcriptional and epigenetic programs, an experimentally supported cis-regulatory domain, and epigenetic mechanisms contributing to cell type-specific and sex-specific regulon composition.

We observed inter-animal differences in specific cell types in both snRNAseq and snATACseq datasets. We considered whether these differences might be the result of technical or tissue handling artifacts. Arguing against technical artifacts, the differences were restricted to specific cell types, with other cell types from the same animals being similar. All samples of each sex were processed simultaneously, reducing any laboratory processing variability. Finally, *post-mortem* changes can be excluded because these pituitaries were snap-frozen immediately after collection[45]. These considerations support the interpretation that these datasets have captured animal-to-animal differences in the in vivo transcriptome and accessibility landscapes.

Analysis of the paired sn transcriptome and chromatin accessibility datasets from individual animals suggests the presence of a continuum between developmentally determined cell types and transient cell states. Analysis of our sn multi-omics data identifies and characterizes all the classical major pituitary cell types at the levels of RNA expression, chromatin accessibility, and methylation. We find the most cell clusters in snRNAseq (18 in male, 18 in female), followed by snATACseq (11 in male, 15 in female) and the fewest in sn methylome analysis (9 in male). Our results are consonant with the formulation that major developmental cell type assignment depends on methylation state[46]. All major known pituitary cell types are identified in all three data modalities. However, whereas transcriptome and chromatin accessibility studies of pooled samples cannot unravel whether additional cell clusters are present in all specimens, our dataset has the power to resolve in transcriptome and chromatin

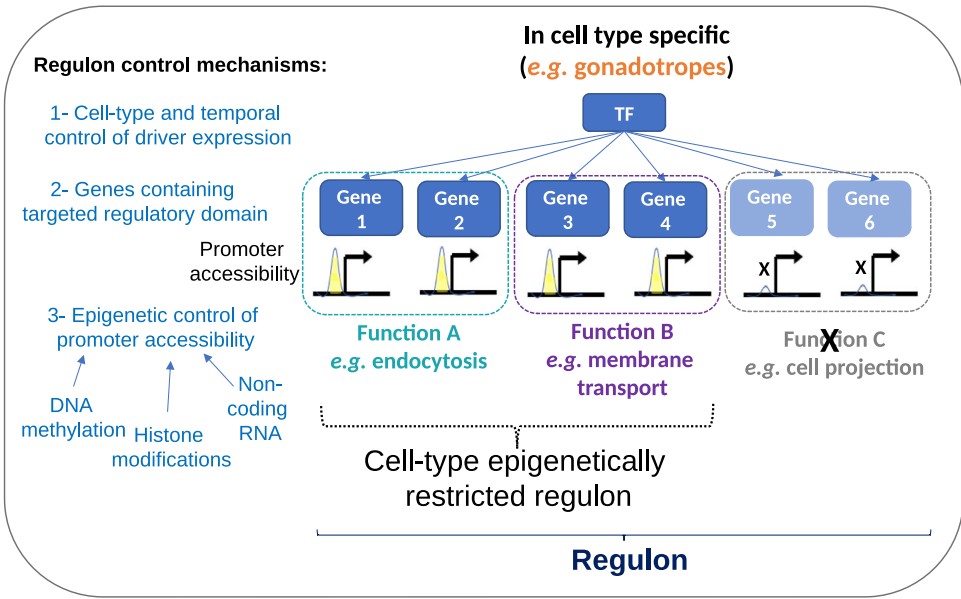

**Fig. 9 Model of cell type-specific regulon control mechanisms.** Schematic of the proposed mechanisms for cell type-specific regulation of regulon composition and function. Three layers of control shape the expression and composition of a cell type-specific regulon and its biological function: cell type-specific expression of the TF driver, presence of cis-regulatory domain at the target gene promoter, and epigenetic control of promoter accessibility.

accessibility data distinct clusters that originate from different animals. For example, analysis of clusters of several cell types seen only with male snRNAseq data identifies these clusters as coming from one male specimen. Their differentiating transcript expression is comprised of IEGs. Therefore, these clusters represent a transient change in cell physiology that we speculatively attribute to some perturbation of this animal prior to death. In females, we observe differences in cell clusters between individuals, which we attribute to one animal likely being in a distinct reproductive (e.g., estrous cycle) state.

Cluster fluidity was also observed in the merged snRNAseq and snATACseq datasets. A somatotrope subcluster closer to lactotropes and most clearly distinguished in the snRNAseq data shared expression of some lactotrope markers in addition to somatotrope markers, expressing a low level of *Prl* and a relatively high level of *Gh* (Fig.2a, Supplementary Fig. 2a and 4a). Previous histological studies have described subpopulations of somatotropes in rat and pig[47–49], including cells co-expressing *Prl* and *Gh*[50–52]. Recent scRNAseq analysis of murine pituitaries also resolved somatotrope sub-populations[1,53]. Our analysis raises the possibility that what has been identified as a somatotrope subcluster might represent an intermediate cell type between the somatotrope and lactotrope lineages.

We note several limitations in the present analysis of cell types and cell states. This study is based on the analysis of data from a small number of animals, and further investigation on larger numbers of samples is warranted. Another caveat is that the snMethyl analysis used pooled samples and was performed only in males, which provides no methylation-level data on inter-animal variation and no information on females. As high-quality data were obtained from only a few thousand nuclei, the resolution of the methylation data for cell subpopulations is limited. Despite these caveats, these results point the way towards a refined model of cell identity. We propose the working hypothesis that cell state and cell type form a continuum gradually shifting from transient transcriptional changes to more stable developmentally programmed changes in expression, chromatin structure, and methylation. Importantly, our results provide the foundation for future studies of correlated transcriptional and epigenetic changes across the reproductive cycle and in response

to perturbations, having the statistical power furnished by individual subject resolution.

We identified distinguishing transcriptional and chromatin accessibility programs for each major cell type. Using the PLIER framework, we determined coordinated cell type-specific gene sets showing inter-sexual and inter-individual differences in RNA expression and promoter chromatin accessibility (Fig. 7, Supplementary Fig. 21). Distinctive gene expression and promoter accessibility patterns were found in females, and cell type-specific patterns were distinguished across individuals, corresponding to the proposed differences in reproductive state among the females. Since these animals were randomly selected with no determination of cycle status, further studies are needed to precisely define the relationship between reproductive endocrine status and coordinated gene program changes in the pituitary. Earlier studies identified a sexual difference in transcriptome-wide gene expression in the pituitary gland[54], and recently in scRNAseq analysis of the mouse pituitary[3]. Similar to Ho et al., we observed a predominance of somatotropes in males vs. females, and a higher proportion of lactotropes in females vs. males. Understanding the sexual differences in the pituitary gland within both transcriptomic and epigenomic layers could have important clinical implications in humans, such as for the development of gender-specific treatments for pituitary disorders.

The transcriptional state of a cell is governed by the activity of GRNs or regulons. Regulons are GRN gene sets that share a common TF driver (see Fig. 8a). Notably, when the snRNAseq data were deconvoluted into pituitary regulons using the SCENIC framework, the activity of regulons within each cell was sufficient to reliably cluster the major pituitary cell types (see Fig. 8c). Thus, regulon activity, which reflects the physiology of each cell, enables its classification. Expression of the regulon gene set may be identified in one or more cell types and should require an expression of its TF driver. Consistent with the key regulatory role of TF driver expression, most pituitary regulon TFs are found to be expressed in a single or limited number of cell types (see Fig. 8b).

When a regulon is expressed in more than one cell type, each of the cell types may express only a subset of the entire pituitary-based regulon gene set[40]. The mechanism for this cell

type-specific expression is unknown. To study possible epigenetic mechanisms, we first investigated the FOXP2 regulon, which is expressed in gonadotropes, melanotropes, and stem cells. *Foxp2* is a novel gonadotrope-enriched marker identified both by scRNAseq and by immunohistochemistry in vivo[1]. Its function in the pituitary has not been investigated. The genes comprising the FOXP2 regulon in the three cell types differed significantly (see Fig. 8d). Notably, we showed that promoter chromatin accessibility and expression level of genes comprising the entire FOXP2 are highly correlated, indicating that the chromatin accessibility landscape in each cell type is shaping the cell type-specific regulon composition. We were unable to identify gene methylation features that strongly correlate with gene activity. As regulon expression and genome-wide DNA methylation levels could independently segregate the major pituitary cell types, this lack of correlation of cell-type-specific regulon genes and methylation was surprising. Our results suggest that other mechanisms may regulate the epigenetic landscape that shapes the cell type-specific regulons.

Sex-specific regulons within a pituitary cell type have not been reported thus far. We identified and characterized two sex-specific regulons expressed only in gonadotropes, those driven by FOXL2 and by NR5A1. FOXL2 is essential in basal and activin-stimulated *Fshb* transcription and FSHβ expression in gonadotropes, and subsequently impacts mouse fertility[55–59]. NR5A1 mediates basal expression of *Lhb*, *Fshb*, and *Cga*[60–62]. Nr5a1 knockout mice are sterile, do not mature sexually, and show decreased LH and FSH levels[63]. For each of these two regulons, we found differences in the regulon composition between males and females that are highly correlated with promoter chromatin accessibility within each sex. There was no high correlation between gene methylation and gene expression. Thus, identification of these sex-specific regulons shaped by chromatin accessibility patterns may contribute to elucidating the mechanisms responsible for differences in the physiology and function of male vs. female gonadotropes.

Both the cell type-specific regulon and the sex-specific regulons that we studied are annotated to different functional processes in different cell types or sexes. This functional module analysis indicates that the differing composition of the cell type- and sex-specific regulons may serve distinct biological functions in different cell types in the two sexes, respectively. Based on these findings, we propose a model where TF driver expression, target site presence, and gene promoter chromatin accessibility represent three layers of control that influence the expression and composition of cell type-specific and sex-specific regulons and their physiological roles (Fig. 9). Elucidating the interaction between the epigenetic mechanisms contributing to gene chromatin accessibility and cell type- and sex-specific regulon composition represents an important area for further investigation.

We have developed and presented our datasets and analyses as a resource for the research community (snpituitaryatlas.princeton.edu). As we demonstrate, this resource can be used for analyses ranging from global evaluation of candidate mechanisms underlying GRN activity to the identification of a novel upstream cis-regulatory site in a specific gene of interest. We include comprehensive co-accessibility analysis in this resource. As a test case of this resource for generating specific hypotheses, we identified a co-accessible region upstream of the *Fshb* gene that corresponds to the site of the human SNP rs11031006, and generated experimental support for the regulatory role of this region. We demonstrate the value of integrated multi-omics and single animal datasets in providing deeper insight into the identity of cells and the mechanisms contributing to their distinguishing transcriptional programs. Our work lays the foundation for characterizing the general epigenetic regulatory principles that

control cell type-specific animal-specific and sex-specific gene expression programs.

## Methods

**Animals and pituitary collection.** Pituitaries were collected from male and randomly-cycling C57BL/6 female mice aged 10–12 weeks. Animals were on a 12-hour on, 12-hour off light cycle (lights on at 7 AM; off at 7 PM). Pituitaries were immediately snap-frozen following dissection, and stored at −80 °C.

**Animals and dissociation of primary murine pituitaries**
*Standard protocol used for dissociation.* Pituitaries were collected from male and randomly cycling female C57BL/6 mice aged 10–12 weeks. Animals were on a 12-hour on, 12-hour off light cycle, with lights on at 7 AM and off at 7 PM. Upon collection, single pituitaries were digested with 40 µl of 1.5 mg/ml collagenase (Sigma, Type I-C0130) in low $Ca^{2+}$ HBSS (pH 7.2–7.4) and dispersed with 4 µl of 2.5 µg/ml pancreatin (10X; Sigma #P3292) in 0.85% NaCl with 36 µl HBSS (NO CALCIUM) and the dissociated cells were collected acutely. About $0.7–2.5 \times 10^6$ cells were collected from single pituitaries and placed in individual tubes for cryopreservation.

*Improved protocol used for gentle dissociation with downstream SC assays.* Upon removal, pituitaries were cut in half, rinsed 2x in Ca/Mg-free PBS, and transferred to a tube containing trypsin solution (final concentration at 0.37 mg/ml, Sigma T9935) and gently agitated at 37 °C for 10 min. After gentle aspiration and expulsion of the tissue, the removed supernatant was diluted with MEM/10% FBS, and the process repeated for three cycles. The combined supernatants were centrifuged and the pellet resuspended in MEM/10%FBS. Approximately $1.7–2.5 \times 10^6$ cells were collected from single pituitaries.

**Ethical compliance.** We have complied with all ethical regulations and institutional protocols. All murine work was conducted at McGill University (Montreal, Quebec, Canada) under animal use protocol 5204, as approved by the Facility Animal Care Committee of the Goodman Cancer Research Centre.

**Cryopreservation/thawing of dissociated pituitary cells.** Dissociated pituitary cells were cryopreserved in 90% FBS/10% DMSO and shipped to New York on dry-ice. Upon arrival, cells were stored in liquid nitrogen. On the day of an experiment, the vial was removed from liquid nitrogen on dry-ice, thawed quickly in a 37 °C water bath, and cells were serially diluted with 10% FBS /DMEM until a minimum volume of 8 ml was reached. Cells were centrifuged at 300xg for 5 min, resuspended in 1 ml of medium and counted.

**ScRNAseq assay.** ScRNAseq was performed as described (10x Genomics, Pleasanton, CA; (24)), following the Single Cell 3' Reagents Kits V2 or V3 User Guidelines. Cells were filtered, counted on a Countess instrument, and resuspended at a concentration of 1,000 cells/µl. The number of cells loaded on the chip was determined based on the 10× Genomics protocol. The 10X chip (Chromium Single Cell 3' Chip kit A v2 PN-12036 or v3 chip kit B PN-2000060) was loaded to target 5,000–10,000 cells final. Reverse-transcription was performed in the emulsion and cDNA was amplified before library construction following the Chromium protocols v2 or v3. Each library was tagged with a different index for multiplexing (Chromium i7 Multiplex kit PN-12062). Quality control and quantification of the amplified cDNA were assessed on a Bioanalyzer (High-Sensitivity DNA Bioanalyzer kit). Sequencing was carried out at the NYGC on Illumina Novaseq using 98 + 26 paired-end reads.

**Quality control (QC) assessment.** We tested different protocols to assess the best approach for building a murine sc pituitary atlas. We assessed the quality of our samples and data by measuring cell viability for cryopreserved dissociated pituitary cells, and by observing the nuclei shapes and sizes during nuclei isolation and counts. We followed the standard QC recommended in the 10× Genomics protocols, making sure the emulsion step was not compromised, performing bionalyzer assessment of the samples after the cDNA amplification stage for sc/sn RNAseq, as well as after library preparation for all samples (sc/sn RNAseq and sc ATACseq). After sequencing standard metrics were assessed in Cell Ranger to make sure our samples were passing QC. Sequencing metrics are presented for all samples in supplementary tables and figures.

**Nuclei isolation from dissociated pituitary cells.** Nuclei were isolated from all cells remaining after the scGEM Drop-seq RNA-seq assay was done. Cells were centrifuged at 500 × g for 5 min at 4 C. The supernatant was carefully removed and 0.1X lysis buffer (1 × : 10 mM Tris-HCl pH 7.5, 10 mM NaCl, 3 mM MgCl₂, nuclease-free H₂O, 0.1% v/v NP-40, 0.1% v/v Tween-20, 0.01% v/v digitonin) was added. After 5 min incubation on ice, 1 ml of wash buffer was added. The nuclei were pelleted at 500 × g for 5 min at RT, resuspended either in 1X PBS/0.04% BSA for scGEM Drop-seq RNA-seq or in diluted nuclei buffer for snATAC-seq. Nuclei were counted and the concentration adjusted to run the given assay.

**Nuclei isolation from pituitaries**. On the ice, snap-frozen pituitaries were thawed and prepared based on a modified protocol from[64]. Same-sex samples were processed simultaneously, with all three female samples processed one day, and all three male samples processed another day. Briefly, RNAse inhibitor (NEB MO314L) was added to the homogenization buffer (0.32 M sucrose, 0.1 mM EDTA, 10 mM Tris-HCl, pH 7.4, 5 mM CaCl2, 3 mM Mg(Ac)2, 0.1% IGEPAL CA-630), 50% OptiPrep (Stock is 60% Media from Sigma; cat# D1556), 35% OptiPrep and 30% OptiPrep right before isolation. Each pituitary was homogenized in a Dounce glass homogenizer (1 ml, VWR cat# 71000-514), and the homogenate filtered through a 40 μm cell strainer. An equal volume of 50% OptiPrep was added, and the gradient centrifuged (SW41 rotor at 17,792 × g; 4 C; 25 min). Nuclei were collected from the interphase, washed, resuspended either in 1X nuclei dilution buffer for snATACseq (10× Genomics) or in 1X PBS/0.04% BSA for snRNAseq, and counted (Invitrogen Countess II).

**snRNAseq assay**. SnRNAseq was performed following the Single Cell 3' Reagents Kits V3 User Guidelines (10x Genomics, Pleasanton, CA). Nuclei were filtered and counted on a Countess instrument. A minimum of 1,000 nuclei were loaded (Chromium Single Cell 3' Chip kit A v2 PN-12036 or v3 chip kit B PN-2000060). Reverse-transcription (RT) was performed in the emulsion, cDNA amplified, and library constructed with v2 or v3 chemistries. Libraries were indexed for multiplexing (Chromium i7 Multiplex kit PN-12062). Same-sex samples were processed together from nuclei isolation to sequencing, with the three female samples processed one day, and the three male samples processed another day.

**snRNAseq data analysis of individual samples**. SnRNAseq data were processed using the Cell Ranger pipeline v3.0.2, and aligned to whole transcripts rather than exonic regions only so as to include pre-mRNA. This resulted in a 2.15-fold increase in the median number of genes detected per cell in females and $2.3 - 2.4$ fold increase in males relative to a traditional exonic-only alignment. Clustering and differential gene expression analysis were performed using Seurat v.3.1.1 and standard procedures[65,66]. SnRNAseq data contain a background level of the most expressed genes in the sample. We first identified gel-beads in emulsion (GEMs) that contained only background as a peak in the UMI count distribution around 255 counts. We used troughs at both the high-end and low-end of the background peak to delimit the range of UMI counts to include in our background calling. Typically, this gave us a UMI counts range of roughly 150 to 400. GEMs that include cells were called with a minimum UMI count of 500 in Cell Ranger, and were thus well separated from those consisting of pure background. We verified that all expressed genes correlate linearly with the total UMI count within background gel-beads, meaning that their proportions did not change. Next, we added 200 random background barcodes into the output Cell Ranger cell-filtered feature matrices, and repeated the clustering. Background GEMs all clustered together and separated from any of the cell clusters (except for the debris cluster), confirming their homogeneous and distinct nature.

We used our background identified GEMs to derive the background distribution of each expressed gene. While only a handful of transcripts were present at a concentration larger than one molecule per gel-bead, they could nevertheless reach counts of up to ~100 in the case of *Gh* in male samples and *Prl* in female samples. Transcripts with a background concentration larger than 0.5 per gel-bead in any one sample were excluded from the clustering analysis. We identified doublets following a specific step-by-step approach (Refer to Supplementary Information for the full description, Supplementary Figs. 28-31). All feature plots for individual samples (UMI counts, mitochondrial content, and ribosomal gene contents) are presented in Supplementary Figs. 32-34.

**snATACseq assay**. SnATACseq was performed following the Chromium Single Cell ATAC Reagent Kits V1 User Guide (10x Genomics, Pleasanton, CA). Nuclei were counted (Countess counter), transposition was performed in 10 μl at 37 C for 60 min on at least 1,000 nuclei, before loading of the Chromium Chip E (PN-2000121). Barcoding was performed in the emulsion (12 cycles) following the Chromium protocol. Libraries were indexed for multiplexing (Chromium i7 Sample Index N, Set A kit PN-3000262). Same-sex samples were processed together from nuclei isolation to sequencing, with the three female samples processed one day, and the three male samples processed another day.

**snATACseq analysis of individual samples**. SnATACseq data were processed using Cell Ranger-ATAC pipeline version 1.2.0 that eliminates barcode multiplets. Same-sex samples were combined using the Cell Ranger-ATAC *aggr* function. We used Signac version 1.0.0 to perform clustering analysis of the merged samples, and used the pipeline's intrinsic *K-Means* clustering analysis for individual samples, as the pipeline's graph-based clustering tended to over-cluster. We employed a $k = 10$ (females), and a $k = 7 - 9$ (males), as they had fewer sequenced cells. Cell types were annotated based on cutsite pileup sums at promoter regions of known cell type markers. Cell types present in too low numbers for the *K-Means* algorithm (which is limited to $k = 10$) were identified as a separate cluster (for example non-pituitary cells). Nevertheless, we were able to visually locate these less abundant cell types as split on the tSNE plot and identify them using promoter sums of known

markers. In individual male samples, which contain fewer nuclei, we were unable to split non-pituitary cells apart since their numbers were too low.

**Doublet identification in the snRNAseq datasets**. Same-type doublets/multiplets were mitigated by setting an upper limit UMI at 45,000 counts. Cross-type doublets were identified in two ways described as follows. Doublets of the most abundant cell types formed separate clusters and were identified by our clustering analysis. This was the case for Somatotrope-Lactotrope doublets as well as Melanotrope doublets and Stem cell doublets (with other cell types). Supplementary Fig. 28a shows the results of our clustering analysis on a t-SNE projection in which we identified four clusters as doublets. Supplementary Fig. 28b shows the same t-SNE plot color-coded by total UMI count and illustrates the higher UMI counts of those barcodes populating doublet clusters. Supplementary Fig. 28c shows a gene expression heatmap of the Melanotrope doublets cluster. Shown are the top-20 markers of some of the main cell types. We could clearly recognize the expression of the top melanotrope markers in this cluster, but in addition, we saw the expression of other gene programs. Based on which of these gene programs were expressed in any given barcode, we were able to infer the exact two (or more) cell types constituting each doublet (or multiplet). Supplementary Fig. 28c is organized in such a way as to illustrate this, as we grouped barcodes by exact cell-type doublet/multiplet. All barcodes that form this cluster were thus confirmed to be indeed doublets or multiplets.

The second step of the doublet identification was motivated by the realization that those doublets which were not numerous enough to form their own cluster at a resolution that makes sense for the sample at hand, will be amalgamated into one or the other of the main cell type clusters. Nevertheless, these would show up on a heatmap of gene expression as shown in Supplementary Fig. 29, which illustrates the gene expression of the initial Corticotropes cluster and where many barcodes could be observed to display expression of a secondary program (showing up as "streaks" on the heatmap). These could be flagged and given an identity that reflects their doublet or multiplet gene expression. The expression in the cleaned sample is much more homogeneous, and, in particular, free of the streaks observed in the original clustering. Running through the heatmap of all of the original clusters, we were thus able to flag and identify all doublet and multiplet barcodes.

We note that the level of expression of a secondary program can sometimes be significantly lower than that of the primary expression. More often than not, the two cells constituting a doublet will not contribute equally to the signal. Nevertheless, the signal will be ubiquitous across the set of markers of the secondary cell type, and this is what we looked for. To make our identification, we used the entire list of markers not only the top-20. Doublets with a dominant cell type would cluster with and project onto the space occupied by cells of that dominant type, whether on a t-SNE or UMAP. They nevertheless stood out on the gene expression heatmap and could be properly identified.

As a third step, we verified our doublet identification by running through all of the two main cell type combinations and generated plots such as those shown in Supplementary Fig. 30, which illustrates, in the case here of Melanotropes – Lactotropes1-2 doublets, how doublets break away from one or the other of their constituting cell types in the aggregate expression of cell-type markers. In addition, we could see that doublets displayed higher UMI counts (Supplementary Fig. 30d), further validating their identification.

Lastly, we looked at the UMI count distribution as a function of cell quality, or multiplet order (Supplementary Fig. 31) Despite the fact that we identified doublets, triplets and higher-order multiplets solely based on their gene expression, we observed, as expected, a regular increase in the UMI count from single-cell barcodes to doublets, to triplets and to clumps (higher-order multiplets). We also noticed a broadening of the distributions, which is also an expected behavior of combinatorial probability. In addition to multiplets, Supplementary Fig. 31 also shows that barcodes identified as containing background (ambient RNA), debris, or low-quality / damaged cells. These pile up at the low cutoff limit of UMI counts. Finally, barcodes in which more than 0.5% of transcripts were hemoglobin transcripts were flagged as contaminated with erythrocyte debris.

**Merged snRNAseq or snATACseq analysis**. All male and female samples were merged by sex in Seurat at the UMI count level, and all of the clustering analysis was repeated on the merged samples independently from the beginning. We followed the same analysis steps as for individual samples. Unlike our integrated samples (see later section), the merged samples do not have batch effects removed. Despite that, we do not observe any systematic batch effect between our samples. We do, however, see differences in gene expression from one animal to another among some specific cell types. The merged samples allow us to highlight these differences in the implicated cell types.

**Quality control (QC) and sequencing of libraries**. QC and quantification of libraries were done by Bioanalyzer (High-Sensitivity DNA Bioanalyzer kit), Qubit (Thermofisher), and Mi-seq (Illumina). Sequencing was carried out at the New York Genome Center (NYGC) on an Illumina Novaseq using $98 + 26$ paired-end reads.

**snMethyl assay**. The snmC-seq2 protocol was applied to individual nuclei isolated from a pool of 30 adult male pituitaries, following a protocol that included bisulfite conversion and subsequent preparation of the snmC-seq2 libraries as previously described[25,67]. Sequencing was done at the Salk Institute on Illumina Novaseq 6000 using 150 bp paired-end reads. Female pituitaries were not analyzed by snMethyl due to the heterogeneity caused by cycling, and the technical requirement of pooling pituitaries.

**snmC-seq2 mapping and preprocessing**. A versatile mapping pipeline (cemba-data.rtfd.io) was implemented for all the sc methylome-based technologies developed by the Ecker Lab[25,67,68]. The main steps of this pipeline included: (1) demultiplexing FASTQ files into sc; (2) reads-level QC; (3) mapping; (4) BAM file processing and QC; (5) final molecular profile generation. Details of the five steps for snmC-seq2 were described previously[33]. We mapped all the reads onto the mouse mm10 genome. After mapping, we calculated the methyl-cytosine counts and total cytosine counts for two sets of genomic regions in each cell. The non-overlapping 100 kb genomic bins of the mm10 genome (generated by "bedtools makewindows -w 100000"), which was used for clustering analysis and ANN model training, and the gene body region ± 2 kb defined by the mouse GENCODE vm22, which was used for cluster annotation and integration with the other modalities.

**Clustering analysis and DMR calling**. Before clustering, we first filtered the cells based on these main mapping metrics: (1) mCCC rate < 0.03. mCCC rate reliably estimates the upper bound of bisulfite non-conversion rate[67], (2) overall mCG rate > 0.5, (3) overall mCH rate < 0.2, (4) total final reads > 500,000, (5) bismark mapping rate > 0.5. The clustering analysis was performed as previously described[68]. In brief, we used the mCG and mCH 100 kb genomic bins matrices as inputs of the clustering analysis. After calculating the normalized posterior estimation of mCG and mCG rate based on the beta-binomial distribution, we selected the top 2500 highly variable features and performed Principal Component Analysis (PCA) on each matrix, and concatenate the first 20 PCs from both matrices. We performed KNN-Louvain clustering (K = 25, Louvain resolution = 0.8) on the concatenated PCs. We manually annotated clusters based on marker genes learned from other modalities. After clustering, we merged the sc ALLC files (whole-genome cytosine count table for each cell) by cluster to get pseudo-bulk ALLC files. We then performed differential methylated regions (DMR) analysis using the "methylpy DMRfind" function among all clusters, as previously described[68]. For cell-type identification, only mCG levels at the promoters of known pituitary cell type genes were used as non-CG DNA methylation (mCH) was a rare event in our pooled male sample.

**PLIER data analysis of sc and sn assays**. PLIER analysis was performed on scRNAseq studies done on dissociated pituitary cells from nine samples (four females and five males) applying the latent variables (LVs) obtained from the snRNAseq data from the snap-frozen samples.

B is a PLIER-derived expression value for the genes associated with a given LV across the different samples. It can be treated similar to average expression, weighted by gene association with the LV. Technically, B is a matrix of size #LVs x #Samples. It is one of two matrices in PLIER, along with Z of size # of genes x #LVs. The goal of PLIER is to find values of B and Z that minimize the equation $||Y - Z*B||$ where Y is our data matrix of size #genes x #samples. So PLIER is finding a suitable number of LVs that can be used to connect the genes and samples and accurately estimate our data matrix.

To examine more deeply the trends in gene expression of assigned cell types across samples and data types, we treated each sc dataset as a collection of bulk datasets for given labeled cell types. Each cell type was then treated as a separate bulk measurement within each sample.

For snATACseq data, peak counts for a given gene were generated by selecting the peak closest to the transcription start site (TSS). These peak counts per gene were then collected into single bulk measurements for each cell type in each sample. We focused specifically on eight relevant cell types in the pituitary: corticotropes, gonadotropes, lactotropes, melanotropes, proliferating cells, somatotropes, stem/progenitor cells, and thyrotropes. For the snRNAseq dataset, this process generated 48 bulk measurements over six samples (three females and three males), and for the snATACseq dataset, we generated 42 bulk measurements. Proliferating cells were not identified in snATACseq. We applied PLIER[37], which finds patterns in count data that are associated with known prior information (such as Reactome and Kegg). PLIER was run on each set of samples separately with LVs generated on the bulk measurements in an unsupervised fashion. LVs were then curated to find patterns relevant to individual cell types as well as sample-wide trends such as sex-based differences. Statistical significance of LVs was computed through the Kruskal–Wallis non-parametric test for multiple groups as part of the stat_compare_means R method. Comparisons between LVs within and across datatypes were achieved by comparing the overlap of the 50 genes most associated with a given LV.

**Generation of the CRISPR LβT2 clone**. The murine LβT2 cell line[69,70] was a gift from Dr. Pamela Mellon (UCSD). Cells were maintained in 10-cm diameter dishes

containing Dulbecco's Modified Eagle's Medium (DMEM) supplemented with 10% fetal bovine serum (FBS) and 1% penicillin/streptomycin (Invitrogen, Carlsbad, CA) in a humidified incubator at 37 °C and 5% $CO_2$. A 20 bp target site GCCCTGTGATATTTATTTCA was chosen to induce double-strand DNA breaks at the TT location using CRISPR-Cas9 (University of Utah Mutation Generation and Detection Core). LβT2 cells were transfected at 50% confluency with 4 μg CRISPR Cas9-gRNA-GFP constructs, and 1 μg of a gel-purified mutagenic primer targeting mouse rs11031006 using PolyJet in vitro DNA Transfection Reagent (SignaGen Laboratories). At 18 h post-transfection, cells were washed with PBS, digested with 0.5 ml 0.25% trypsin-EDTA, and reconstituted with 0.5 ml DMEM. Cells were sorted for GFP expression and collected by an Avalon Cell Sorter (Propel Labs, Fort Collins, CO), before being plated on a 12-well plate in supplemented DMEM. Isolation, genotyping, and expansion of individual colonies were repeated three times to establish new clonal cell lines. The QIAamp DNA Mini Kit (QIAGEN, Hilden, Germany) was used to extract DNA of each colony for verification of the clonal genotype by Sanger sequencing using rs11031006 primers. Genotype was identified using Poly Peak Parser[71]. One clone was selected for further studies.

**Quantitative real-time PCR (qPCR)**. Edited and unedited LβT2 cells were plated at 1,500−5,500 cells per well into 96-well plates in DMEM with 10% fetal bovine serum and 1% penicillin/streptomycin. Cells were treated for 24 h either with vehicle or with 10 nM GnRH and 1 nM activin A (R&D Systems, Minneapolis, MN) in serum-free DMEM with 0.1% BSA.

Total RNA from each well was extracted using the RNeasy Mini Kit (Qiagen, Valencia, CA) and reverse transcribed to cDNA using SuperScript VILO MasterMix (ThermoFisher, Rockford, IL). Samples were then subjected to qPCR using PowerUp SYBR Green Master Mix (ThermoFisher, Rockford, IL). qPCR reactions were carried out with the following conditions: 50 °C for 2 min; 95 °C for 10 min; 40 cycles of 95 °C for 15 sec and 60 °C for 1 min using an Applied Biosystems 7900HT (Foster City, CA). SDS 2.1 software was used to identify the cycle number for each target (Applied Biosystems, Carlsbad, CA). Three technical qPCR replicates were run for each biological replicate. Results were exported as cycle threshold (Ct) values, and Ct values of target genes were normalized to those of Gapdh in subsequent analysis. Data were expressed as arbitrary units by using the formula, $E = 2500 \times 1.93^{(Gapdh\ CT\ value\ -\ gene\ of\ interest\ CT\ value)}$, where $E$ is the expression level in arbitrary units. The data were expressed as the mean ± s.e.m, and a value of $p < 0.05$ was considered statistically significant. All primers used in CRISPR and in qPCR experiments are provided in Supplementary Table 8.

**Radioimmunoassays**. FSH secreted in the medium was assayed, and the reportable range for the FSH assay was 1.1–54.1 ng/mL. The intra-assay and inter-assay %CV for FSH was 7.4% and 9.1%, respectively. These assays were performed at the University of Virginia Core Ligand and Assay Laboratory.

**sn data integration**. The snRNAseq, snATACseq, and snMethyl data were integrated in a reference-query based manner, mainly using the "FindTransferAnchors" and "TransferData" functions from the Seurat v3 package[65,66]. The snRNAseq datasets were used as the reference and the other modalities were integrated into them. To integrate snATACseq to snRNAseq, the peak-by-cell accessibility matrix was converted to a gene-by-cell activity matrix based on the chromatin accessibility within each gene's gene body and a 2 kb upstream region, under the assumption that chromatin accessibility and gene expression were positively correlated. The variable features from the snRNAseq data were used to find the anchors and the snATACseq data in the LSI low-dimensional embedding were used to transfer the data from snRNAseq to snATACseq. To integrate the snMethyl data to snRNAseq, the normalized methylation rates for each gene's gene body and + /− 2 kb flanking region were used. This gene-by-cell methylation rate matrix was converted to an activity matrix by "activity = 1 / normalized methylation rate", under the assumption that methylation and gene expression were negatively correlated. The variable features from the snRNAseq data were used to find the anchors and the snMethyl data in the PCA low-dimensional embedding were used to transfer the data from snRNAseq to snMethyl. Co-embedding of the three modalities was done by concatenating snRNAseq with the transferred data of snATACseq and snMethyl, and running dimension reduction on the concatenated dataset.

**Regulon analysis**. We used SCENIC[39] to reconstruct the GRNs from snRNAseq data, with 3 steps: (1) identifying sets of genes that are co-expressed with TFs in the snRNAseq dataset, (2) refining the target genes for each TF based on enrichment of its cis-regulatory binding motifs, and (3) measuring the enrichment of each regulon (consisting of a TF and its target genes) in each nucleus using an AUC score (regulon activity score). SCENIC was run on filtered and pooled datasets of three females and three males with the raw UMI counts used as an input. In each dataset, only genes with UMI count > = 3 in at least 3 cells were kept, and cells identified as doublets were removed.

The output was a regulon-by-cell matrix composed of the activity of each regulon in each cell type. Cell type-level regulon activities were calculated by the average of sn-level regulon activities. For a regulon in a cell type, the relationship

between chromatin accessibility and RNA expression of target genes was determined using their relative levels, defined as the ratio between cell type average and whole-dataset average. Chromatin accessibility of the target genes was measured by the accessibility either in the promoter region (500 bp region in the upstream of TSS) or in the gene body region. We determined the relationship between relative gene accessibility and expression by scatter plots and linear regression.

Determination of "relatively high" and "relatively low" expression was done by setting a threshold for defining relatively high and low expression. In the analysis combining multiple samples (e.g., all males), a gene is identified as "high" if it is "high" in at least one sample. To study the target genes specifically turned on in each regulon and each cell type, we identified a target gene as specifically highly expressed in a cell type if it satisfied either of these two conditions: (1) its relative averaged normalized RNA expression in this cell type vs. the whole dataset was greater than 2, and it had non-zero UMI counts in at least 10% of the cells in this cell type; and (2) it had non-zero UMI counts in at least 50% of the cells, and its relative averaged normalized RNA expression in this cell type was greater than 1. The rationale here was to avoid genes being classified as "high" if they were universally lowly expressed across all cells but showed high relative expression because of noise. In the analysis involving multiple samples such as all males, a target gene was defined as specifically highly expressed in a cell type if it was specifically highly expressed in the cell type in at least one sample.

**Regulon activity calculations.** Regulon activity scores for each cell were calculated using the AUCell function from the SCENIC pipeline[39]. Basically, the activity of a regulon in a cell was evaluated by the enrichment of the regulon genes in the highly expressed genes of this cell, and the enrichment was measured by the area under the curve (AUC) of recovering the regulon genes from the expression-based ranking of all genes. The cell-type level regulon activity scores were calculated by averaging the sc level regulon activity scores across all the cells of the cell type.

**Co-accessibility and putative regulatory region.** Co-accessibility between all pairs of snATACseq peaks within 500 kb was calculated using the Cicero package[31]. To find putative distal regulatory regions for a gene, we used the snATACseq peaks within the gene promoter region, and identified their co-accessible peaks that were at least 5 kb away.

**Functional module analysis of target genes.** To identify the functions enriched in the lists of on and off genes in each regulon, we used the functional module detection method from the HumanBase resource (https://hb.flatironinstitute.org/module). The method clusters genes by their connectivities in a tissue-specific functional network and finds enriched GO terms for each of the gene clusters[44].

**Conservation analysis.** PhastCon and PhyloP(60) analyses were performed to analyze conservation among vertebrate species[34,35]. PhyloP was performed on 60 vertebrate species.

**Boxplots.** For all box plots comparing ATAC or Methyl between lowly and highly expressed genes, the box plots are plotted using the quantiles described below. Specifically, the geom_boxplot function from the R package ggplot2 is used. This applies to Supplementary Fig. 22d, Supplementary Fig. 24a,c, Supplementary Fig. 26b,c, and Supplementary Fig. 27a. The lower and upper box bounds correspond to the first and third quartiles (the 25th and 75th percentiles). The upper whisker extends from the upper box bound to the largest value no further than 1.5 * IQR (inter-quartile range, or distance between the first and third quartiles) from the upper box bound. The lower whisker extends from the lower box bound to the smallest value at most 1.5 * IQR from the lower box bound. Data points outside of the range of whiskers are plotted as individual points.

Two-sided Wilcoxon rank-sum tests (a.k.a Mann-Whitney U tests) were used, and p values were calculated by normal approximation. Specifically, the R core function wilcox.test was used.

ggboxplot generates a boxplot with the center equal to the 50th percentile, the bounds of the box are the 25th and 75th percentile and the bounds of the whiskers are the smallest/largest values 1.5 times the interquartile range below the 25th percentile or above the 75th percentile, respectively.

**Statistics.** For the assessment of sex difference in cell type proportions (Fig. 4e, f), we used a two-way analysis of variance (ANOVA) followed by Bonferroni multiple comparisons post-hoc test, with $n = 3$ biological replicates per cell type, $F(11, 47) = 20.80$ for snRNAseq, and $F(8, 15) = 23.77$ for snATACseq.

For analysis of LV sex differences by PLIER (Fig. 7b, Supplementary Fig. 21a), we used a two-tailed Wilcoxon rank-sum test with $n = 48$ (8 cell types, 6 biological samples), 1 degree of freedom, $W = 490$, $p = 1.108e-05$ for the snRNAseq data (Left panel) and with $n = 12$ (2 cell types, 6 biological samples), 1 degree of freedom, $W = 36$, $p = 0.002165$ for the lactotrope and somatotrope snATACseq data (Right panel). For evaluating the cell type differences of gonadotrope-specific LVs by PLIER (Fig. 7b, Supplementary Fig. 21b), we used a two-tailed Kruskal–Wallis analysis to test whether all samples originated from the same

distribution. For LV2rna applied to snRNAseq data (Left panel), $n = 48$ (8 cell types, 6 biological samples), 7 degrees of freedom, $\chi^2 = 34.561$, $p = 1.352e-05$. For LV2atac applied to snATACseq data (Center panel), $n = 42$ (7 cell types, 6 biological samples), 6 degrees of freedom, $\chi^2 = 37.34$, $p = 1.512e-06$. For LV2rna applied to snATACseq data (Right panel), $n = 42$ (7 cell types, 6 biological samples), 6 degrees of freedom, $\chi^2 = 30.813$, $p = 2.752e-05$.

For analysis of LV sex differences by PLIER applied to the dissociated pituitary dataset (Supplementary Fig. 17), we used a two-tailed Wilcoxon rank-sum test with $n = 48$ (8 cell types over 6 biological samples), 1 degree of freedom, $W = 490$, $p = 1.108e-05$ for the snap-frozen snRNAseq data (Left panel—as seen in Supplementary Fig. 21a) and with $n = 64$ (up to 8 cell types over 9 biological samples), 1 degree of freedom, $W = 905$, $p = 1.142e-08$ for the dissociated pituitary RNAseq data (Right panel).

For determining whether the association of PLIER LVs to known prior knowledge gene pathway information was random, we applied a cross-validation test[37] that determines if each association between LV and prior knowledge pathway was robust to removing random one-fifth subsets of the genes associated with each pathway. AUC values were calculated based upon the ability to recover each pathway for a given LV.

For all the boxplots comparing chromatin accessibility or methylation between highly and lowly expressed genes, two-sided Wilcoxon rank-sum tests (a.k.a. Mann–Whitney U tests) were used, and p values were calculated by normal approximation. Specifically, the R core function wilcox.test was used. The number of genes tested and the W values from the Wilcoxon rank-sum tests were: 235 highly vs. 75 lowly expressed genes (W = 3388, Supplementary Fig. 22d), and 669 highly vs. 957 lowly expressed genes (W = 158752, Supplementary Fig. 24a). The number of genes tested and the W values from the Wilcoxon rank-sum tests were: in Supplementary Fig. 26b, 232 highly expressed vs. 75 lowly expressed genes (W = 8995); in Supplementary Fig. 26c, 25 highly expressed vs. 25 lowly expressed genes (W = 175); in Supplementary Fig. 27a, 284 highly expressed vs. 112 lowly expressed genes for males (W = 6771), and 332 highly expressed vs. 64 lowly expressed genes for females (W = 4419); in Supplementary Fig. 24c, 871 highly expressed vs. 755 lowly expressed genes (W = 182337).

For *Fshb* expression (Fig. 6d), statistical analyses were all performed using GraphPad Prism version 5.04 (GraphPad Software, San Diego, CA, www.graphpad.com). We used a two-way ANOVA followed by Bonferroni multiple comparisons post-hoc test, with $n = 4$ biological replicates per sample (each measured in technical triplicates), $F(1, 29) = 29.54$ in the GnRH + activin A treatment in the unedited vs. CRISPR clones, and $F(1, 29) = 22$ in the CRISPR clone for the basal vs. GnRH + activin A treatment.

**Reporting summary.** Further information on research design is available in the Nature Research Reporting Summary linked to this article.

## Data availability

The datasets (scRNAseq, snRNAseq, snATACseq) and snMethyl data generated in the present study are deposited in GEO (GSE151962 and GSE152011). The sn mouse pituitary multi-omics atlas can be browsed via a web-based portal accessible at snpituitaryatlas.princeton.edu. All data is available from the authors upon reasonable request.

## Code availability

Any computational code used in the paper is available upon request.

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

## Acknowledgements

This work was supported by funding from the National Institute of Health (NIH) Grant DK46943 (SCS), the Canadian Institutes of Health Research (CIHR) Project Grants PJT-162343 (DJB) and PJT-169184 (DJB), NIH award R01HD065029 from the Eunice Kennedy Shriver National Institute Of Child Health & Human Development (CKW), NIH NICHD R01HD093461 (GVC), NIH R01HD087057 (GVC), and NIH NIDDK 1R01DK113776-01 (GVC). We acknowledge the Mutation Generation and Detection Core for reagents (University of Utah), the Flow Cytometry Core (University of Utah), the Genomic Core (University of Utah), and the New York Genome Center for sequencing. Radioimmunoassays were performed at the University of Virginia Core Ligand and Assay Laboratory. This work was supported in part by the computational and data resources and staff expertise provided by Scientific Computing at the Icahn School of Medicine at Mount Sinai. J.R.E is an Investigator of the Howard Hughes Medical Institute.

## Author contributions

F.R.Z designed and performed research, analyzed and interpreted data, drafted the paper; Z.Z., M.Z., G.R.S., and G.N. contributed analytic tools and analyzed data; H.L. performed research and analyzed data; M.M., R.G.C., N.M., V.D.N., N.S., M.A.A., X.Z., L.O., G.S., C.T., J.R.N., A.B., A.A., and N.J. performed research; H.P. drafted the paper; J.L.T. analyzed and interpreted data; D.J.B. analyzed and interpreted data and provided analytical tools; C.K.W. designed and performed research, analyzed and interpreted data; O.G.T. and J.R.E. contributed analytic tools; G.V.C. discussed the results; S.C.S. conceived research, analyzed data, and drafted the paper. All authors edited the paper and approved its final version.

## Competing interests

The authors declare no competing interests.
