## [Peer Review File · Nature Communications]

REVIEWER COMMENTS

Reviewer #1 (Remarks to the Author):

This paper presents a comprehensive analysis of the transcriptome, DNA accessibility and methylome in single cells and nuclei of the mouse pituitary gland. Beyond comparing data obtained from single cells with single nuclei and different methods to obtain these samples, the analyses were performed on individual animals of both sexes and they provide a fair assessment of variability arising from individual samples and/or sex. Collectively, the large number of analyzed nuclei provide a thorough and hopefully definitive assessment of the bulk cell composition of the pituitary gland. The most thorough distinction between cell types is provided by the transcriptome data followed by the single nucleus ATACSeq and finally by the CpG methylome data: all 3 parameters converge on similar cell distributions. Despite the fact that the ATAC data interpretation relies on the entire datasets of peaks (the bulk of which likely are enhancers) to differentiate cell types, their validation is mostly focused on promoter accessibility and these are correlated with promoter demethylation. Nonetheless, the data is there and available to assess enhancer/promoter relationships and indeed the authors do pursue this quest with one example/locus, the FSHB locus. Interestingly, a putative enhancer is identified in the FSHB locus and this enhancer corresponds to a single nucleotide polymorphism in humans that was previously associated with fertility. The authors go on to mutate sequences around this site by CRISPR and found that the mutated LHbeta2 cells gain FSH responsiveness to GnRH combined with Activin.

Although four or five papers have already appeared with single cell RNASeq analyses of the mouse pituitary, the present work is the most comprehensive and provides the added value of including DNA accessibility (ATAC) and methylome data. This thorough and comprehensive compendium will be extremely useful to scientists in the field of pituitary endocrinology. Technologically, the analyses appear to be state-of-the-art and well supported.

The FSHB locus analysis taken to support the biological relevance of the atlas (which will be publically available) is however not as thorough. Indeed, the causal relationship between the CRISPR mutation and the gain in GnRH+activin responsiveness needs to be better established. CRISPR mutagenesis may produce off-target effects and selection of a CRISPRed cell line may be subject to clonal discrepancy. A number of complementary experiments could be performed to solidify conclusions based on the mutated LHbeta2 cells. First, it could be assessed whether the mutated locus has enhancer activity and whether the activity of this enhancer is sensitive to GnRH+activin, be it the wild-type or the mutant forms of this putative enhancer. Second, ATAC experiments could be performed on the parental and CRISPR mutant cells to assess whether DNA accessibility of the putative enhancer is altered by the mutation in basal and/or GnRH+activin conditions.

Minor comments

L.425-426. The proposal that "identification of a specific cell lineage as opposed to a transient cell state requires confirmation by..." is strong and in this reviewer's opinion, not really supported by the paper. The word "requires" is in particular objectionable as the transcriptome data on their own suffice and provide the deepest clustering between the different cell types. Furthermore, previous papers on snRNASeq of the mouse pituitary reached similar cell distributions. It is therefore questionable whether the accessibility and the methylome data are "required", notwithstanding the fact that these data add considerably to the depth of the global analyses and will constitute a very useful resource for the field.

L.451 The authors refer to the "importance of FOXI2 in reproductive physiology" with references: for the benefit of unknowing readers, the authors should briefly identify the nature of this "importance".

Reviewer #2 (Remarks to the Author):

In this manuscript, the authors first separately performed snRNA-seq (gene expression), snATAC-seq (chromatin accessibility) and snmC-seq2 (DNA methylation) analyses of mouse pituitary at single cell level. Next, they integrated these single-cell genomics data sets to better understand the cell-type specific gene regulatory networks. Finally, they linked a locus with Fshb gene expression.

The strength of the study is the use of cutting edge single-nucleus multi-omics assays covering both transcriptomic and epigenomic landscapes in mouse pituitaries. But the authors fell short in the data analysis and integration. In its current form, the manuscript lacks essential technical information, and data analysis seems quite preliminary and not very well integrated. The authors should properly address the following concerns regarding data analysis and interpretation.

Major points:

1. For Figure 2b-d and supplementary Figure 3-4, the authors chose to show feature plots, UMI count per cell and mitochondrial gene expression % per cell from one male animal. Generally speaking, snRNA-seq results from different animals should be computationally integrated to generate combined male and female tSNE or UMAP plots (color-grouped by different samples and cell-type annotations). Thus, the batch effects can be properly removed and inter-animal variability evaluated. Seurat3 and other published tools like Harmony have build-in functions to align and integrate the datasets from different samples. Similarly, for Figure 3a and Supplementary Figure 8a, the clustering of different donors should be integrated in one UMAP plot with different samples color-coded.

2. For Fig 2c, there is an obvious effect of different UMI counts on cell clustering. This should be regressed out before PCA analysis.

3. In line 140, the authors stated "Som 0 was unlikely to include doublets because the absolute number of transcripts (UMI counts) was on par with that of the other somatotropes". It is unclear how the authors identify multiplets in snRNA-seq data (no detailed description in the method section). Because of separate clustering from individual animals, it is impossible to evaluate the overall relationship between clusters "Som 0" and "Db" from different animals. In Figure 2a, unlike mouse 1, where "Som 0" and "Db" clusters are well separated, they are quite close in tSNE plots of mouse 2 and 3. Also, it is necessary to show the UMI counts and mitochondrial percentage of these 2 clusters in mouse 2 and 3.

4. The author detected 3 sub-types of somatotropes (Som0, 1, and 2) clusters in snRNA-seq datasets. "Som 0" in Figure 2 appeared to be labeled as "Somato-lactorope" in Supplementary Figure 2, which is consistent with the observation that "Som 0" co-expresses Gh/Sgcz/Prl. The proportion of these sub-types should be summarized as in supplementary Table 2, and the differences in subtypes of somatotypes should be systematically examined and discussed based on the experimental methods (e.g. 3 subtypes of snRNA-seq in Figure 2a, 2 subtypes of snATAC-seq in Figure 3a and 1 cluster of sn methylation data in Figure 3b).

5. For all the reported cell types in Figure 4f, what is the relative percentage of these cell-types in pituitary based on integrated multi-omics datasets? Can the authors identify molecular differences for somatotrope subtypes (0, 1 and 2) in snATAC-seq and snmC-seq2 datasets?

6. In Figure 3, why only male mice were analyzed for single-cell DNA methylation? Which genomic regions (promoter or gene body) are used for calculating the methylation level for marker genes in Figure 3c? Are these CpG or non-CpG methylation? These information are missing in the figure legend and method sections.

7. In Figure 7b-c, many major cell-types (e.g. Somatotropes, Lactotropes) are separated into multiple sub-clusters, which seem to be mainly driven by batch effects rather than gender. For instance, it seems Lac from female #3 is separated away from other two females (#1/2) and is close to male Lac. How did the authors integrate snRNA-seq datasets from different donors in Figure 7c?

8. In figure 9, to support the causal relationship between the point mutations (rs11031006) and Fshb expression, the following experiments should be performed: a) The co-access between the peak/locus and Fshb promoter should be re-examined in the in vitro cell lines first; b) The point mutation (G to A/C/T) should be generated to examine the effect of Fshb expression rather than the deletion of "TATTT" in front of rs11031006.

9. In general, the figures look very preliminary and lack essential information. For instance, in Figure 5, the scale (y-axis) and genomic coordinates are missing for all genomic tracks. In Figure 6, the labels are not explained. What is color-scale of heatmaps? What is the meaning of B-score? The gene names next to the heatmap are too small.

Minor points:

1. The Fig. 2c-d are identical to Supplementary Figure 4 b-c.
2. In line 155-156, Supplementary Figure 8 appears in front of Supplementary Figure 7.
3. In line 157, Supplementary figure 8b does not support the claim "a canonical fragment-size distribution with clearly resolved mono- and multi-nucleosomal modes".
4. In line 161, it should be "30,000 fragments per nucleus" not "30,000 peaks per nucleus".
5. For Supplementary Figure 12, it is necessary to show the scatterplot between methylation and expression of 331 FOXL2 regulon genes as in Figure 7e. Based on the association between gene expression and accessibility as well as between gene expression and TSS methylation, it does not support the strong conclusion that "These results indicate that promoter accessibility is a fundamental determinant of activity of FOXL2 regulon genes within the gonadotrope, whereas methylation has a more limited influence" in line 315-317.
6. Lines 575 and 586, cited wrong reference. Ref#39 is not related to clustering of sn Methylation data sets.
7. For "Mac" clusters in Supplementary Figure 6b, it is better to show some well-known markers of macrophages in the heatmap.
8. In Supplementary Table 1 and Figure S1, the authors described sc/snRNA-seq and scATAC-seq experiments with different sample processing methods (cells vs. Nuclei), tissue dissociation protocols (gentle vs standard) and 10x chemistries (v2 vs. v3). For more meaningful comparisons, the authors should provide more information (#read, #UMI, #gene detected per cell/nucleus, etc.) in Table S1. Also, essential information and quality controls of dissociation protocols should be included in the method section. For example, In the Supplementary Information, the section of "Standard protocol used for dissociation", what is the concentration of collagenase used.

Reviewer #3 (Remarks to the Author):

In this manuscript, Ruf-Zamojski and colleagues present the first large-scale single-nucleus multi-omics atlas of pituitary cells of adult mice. The scope of this work is impressive, including single-nucleus RNAseq, ATACseq, and DNA methylation data from a total of more than 70,000 pituitary cells from both male and female adult mice. By integrating these data, the authors are able to pinpoint differential regulatory gene networks/regulons of genes expressed in multiple pituitary cell-types, such as the transcription factor *Foxp2*, as well as functionally test a gonadotrope-accessible SNP associated with human fertility. Finally, the data have been made openly accessible through an interactive web portal, and these data will be of great interest and utility to the research community.

In identifying gonadotrope regulons of *Foxl2*, is there any functional data to support their role in gonadotrope function or otherwise address the role of these genes and pathways? It is difficult to interpret the biological interpretation of the regulons given the large number of genes and wide variety of biological terms/pathways.

Similarly, can you provide any functional data to suggest how *Foxp2* differentially regulates different genes in different cell-types, e.g. by showing their differential chromatin accessibility or DNA methylation? This data would be strengthened if there were data to functionally test *Foxp2* expression in different pituitary cell lines to demonstrate context-dependent activity.

In Fig 7e, although there is an overall positive correlation between chromatin accessibility and gene expression, there are some highlighted gonadotrope genes that are high accessibility-low expression (*Cga/Fshb*) and vice-versa (*Nr5a1*). What do you make of these outliers? In Fig 7f, what is the definition of 'relatively low' and 'relatively high' expression? In identifying *Foxl2* regulon genes in gonadotropes (Fig 7g/h), could you describe whether the expression of these genes is correlated with high or low expression of *Foxl2* itself?

In Fig 9d, why do the CRISPR deletions not cover the SNP location? Do humans bearing variation at this SNP show increased basal and/or secreted *Fshb* levels? Discuss how increased *Fshb* might relate to PCOS and endometriosis phenotypes in humans.

The amount of data presented can cause the data to be difficult to see in places. Figures 2 and 3 describe the snRNAseq and snATACseq data individually, although they are not subsequently analyzed as separate datasets. In both, why do the expression/accessibility plots display data from mouse 1 only instead of all samples combined? In Fig 3c, the color scale of dark blue to yellow can be difficult to see, particularly for yellow on white. In Fig 5, the peaks under the + and - signs on the DNA methylation data are very difficult to see and there are several that I am unable to discern a difference from other cell-types, e.g. the hypomethylated regions in *Cga* in lactotropes (Fig 5a). The key for Fig 5 may be simpler if ordered top-to-bottom in the same order they appear in the tracks rather than a 2x4 arrangement.

In Fig 6, inference of female mouse 3 as being in non-cycling may be premature given the possibility of individual variety. Could female mouse #3 be in a different estrous cycle stage? Does the data from Boehm on gene expression at different stages of the estrous cycle informative in this regard?

Line 133: typo, SNN is shared nearest neighbor

Line 1065: typo, LV1tacet should be LV1atac

Reviewers' comments are bolded and in italics and the authors' responses follow each comment.

REVIEWER COMMENTS

Reviewer #1 (Remarks to the Author):

This paper presents a comprehensive analysis of the transcriptome, DNA accessibility and methylome in single cells and nuclei of the mouse pituitary gland. Beyond comparing data obtained from single cells with single nuclei and different methods to obtain these samples, the analyses were performed on individual animals of both sexes and they provide a fair assessment of variability arising from individual samples and/or sex. Collectively, the large number of analyzed nuclei provide a thorough and hopefully definitive assessment of the bulk cell composition of the pituitary gland. The most thorough distinction between cell types is provided by the transcriptome data followed by the single nucleus ATACSeq and finally by the CpG methylome data: all 3 parameters converge on similar cell distributions. Despite the fact that the ATAC data interpretation relies on the entire datasets of peaks (the bulk of which likely are enhancers) to differentiate cell types, their validation is mostly focused on promoter accessibility and these are correlated with promoter demethylation. Nonetheless, the data is there and available to assess enhancer/promoter relationships and indeed the authors do pursue this quest with one example/locus, the FSHB locus. Interestingly, a putative enhancer is identified in the FSHB locus and this enhancer corresponds to a single nucleotide polymorphism in humans that was previously associated with fertility. The authors go on to mutate sequences around this site by CRISPR and found that the mutated LHbetaT2 cells gain FSH responsiveness to GnRH combined with Activin.

Although four or five papers have already appeared with single cell RNASeq analyses of the mouse pituitary, the present work is the most comprehensive and provides the added value of including DNA accessibility (ATAC) and methylome data. This thorough and comprehensive compendium will be extremely useful to scientists in the field of pituitary endocrinology. Technologically, the analyses appear to be state-of-the-art and well supported.

The FSHB locus analysis taken to support the biological relevance of the atlas (which will be publically available) is however not as thorough. Indeed, the causal relationship between the CRISPR mutation and the gain in GnRH+activin responsiveness needs to be better established. CRISPR mutagenesis may produce off-target effects and selection of a CRISPRed cell line may be subject to clonal discrepancy. A number of complementary experiments could be performed to solidify conclusions based on the mutated LHbetaT2 cells. First, it could be assess whether the mutated locus has enhancer activity and whether the activity of this enhancer is sensitive to GnRH+activin, be it the wild-type or the mutant forms of this putative enhancer. Second, ATAC experiments could be performed on the parental and CRISPR mutant cells to assess whether DNA accessibility of the putative enhancer is altered by the mutation in basal and/or GnRH+activin conditions.

Authors' responses:

The analysis of the FSHB locus represents a minor point of the study and is included as a use-case example of how the atlas can be used to generate hypotheses for further study. As noted by the reviewer, the point of this study is to provide a “thorough and comprehensive compendium [that] will be extremely useful to scientists in the field of pituitary endocrinology.” While a full study of this locus is outside the scope of this multi-omics atlas, in response to the reviewer’s comments we include in Supplementary Fig.18 additional data from an unedited clone and from two CRISPR clones (5G9A2F AND 6C9D), in addition to the one highlighted in the paper (6D8A), to address the reviewer's main concern about CRISPR off-target effects. The detection of similar FSH enhancing responses to different CRISPR modifications, along with results reported from human genetics, in aggregate strongly support the likely importance of this locus in modulating FSH expression. We have revised the Results to stipulate that the Fshb locus analysis represents a use-case example and to include these additional data. Finally, as now indicated in the Results, another group recently validated our hypothesis¹ following the posting of our study preprint². This corroborates the usefulness of our atlas for identifying putative regulatory regions that have a clinical relevance.

Minor comments

L.425-426. The proposal that “identification of a specific cell lineage as opposed to a transient cell state requires confirmation by...” is strong and in this reviewer’s opinion, not really supported by the paper. The word “requires” is in particular objectionable as the transcriptome data on their own suffice and provide the deepest clustering between the different cell types. Furthermore, previous papers on sncRNASeq of the mouse pituitary reached similar cell distributions. It is therefore questionable whether the accessibility and the

methyome data are “required”, notwithstanding the fact that these data add considerably to the depth of the global analyses and will constitute a very useful resource for the field.

We acknowledge the reviewer’s point that there is debate about what constitutes a cell lineage (cell “type”). Nevertheless, because DNA methylation is a relatively stable epigenetic modification, it provides a reliable means for establishing cell types^{3, 4}. On the other hand, transcriptome and chromatin accessibility analysis can also identify cells that transition between states⁵, e.g. following some perturbation. We thus added the following statement in the Introduction: “Sc epigenetic assays, particularly methyome analysis, provide reliable identification of developmentally determined cell types, distinguishing them from potentially transient cell states that can be observed in transcriptome analyses⁵.”

In our study, while the major pituitary cell types were identified using any of the three omics modalities, the most cell clusters were detected by snRNAseq, followed by snATACseq, and lastly by sn methylation analysis. Our analysis of the paired sn transcriptome and chromatin accessibility datasets from individual animals suggests the existence of a continuum between developmentally determined cell types and transient cell states. For instance, snRNAseq and snATACseq both identified several somatotrope sub-clusters with varying patterns of gene expression and chromatin accessibility; two stem cell clusters were distinguished in the snRNAseq data, vs. a single cluster in the snATACseq data. As we now indicate in the Discussion: “We propose the working hypothesis that cell state and cell type form a continuum varying from transient changes in gene program expression, to more prolonged alterations in gene expression and chromatin structure and, finally, to relatively stable developmentally encoded changes in expression, chromatin structure and methylation. Importantly, our results provide the foundation for future study of correlated transcriptional and epigenetic changes during the reproductive cycle and in response to perturbations having the statistical power furnished by individual subject resolution.”

L.451 The authors refer to the “importance of FOXL2 in reproductive physiology” with references: for the benefit of unknowing readers, the authors should briefly identify the nature of this “importance”.

We appreciate the reviewer’s comment, and accordingly have provided more specifics about Foxl2 in the Discussion: “Foxl2 is essential in basal and activin-stimulated *Fshb* transcription and FSH β expression in gonadotropes, and subsequently impacts mouse fertility^{50, 51, 52}.”

Reviewer #2 (Remarks to the Author):

In this manuscript, the authors first separately performed snRNA-seq (gene expression), snATAC-seq (chromatin accessibility) and snmC-seq2 (DNA methylation) analyses of mouse

pituitary at single cell level. Next, they integrated these single-cell genomics data sets to better understand the cell-type specific gene regulatory networks. Finally, they linked a locus with Fshb gene expression.

The strength of the study is the use of cutting edge single-nucleus multi-omics assays covering both transcriptomic and epigenomic landscapes in mouse pituitaries. But the authors fell short in the data analysis and integration. In its current form, the manuscript lacks essential technical information, and data analysis seems quite preliminary and not very well integrated. The authors should properly address the following concerns regarding data analysis and interpretation.

Major points:

1. For Figure 2b-d and supplementary Figure 3-4, the authors chose to show feature plots, UMI count per cell and mitochondrial gene expression % per cell from one male animal. Generally speaking, snRNA-seq results from different animals should be computationally integrated to generate combined male and female tSNE or UMAP plots (color-grouped by different samples and cell-type annotations). Thus, the batch effects can be properly removed and inter-animal variability evaluated. Seurat3 and other published tools like Harmony have build-in functions to align and integrate the datasets from different samples. Similarly, for Figure 3a and Supplementary Figure 8a, the clustering of different donors should be integrated in one UMAP plot with different samples color-coded.

We recognize the general value of snRNAseq data integration across different conditions, technologies, or species in order to eliminate batch effects^{6, 7, 8}. We should point out, however, that the snRNAseq and snATACseq datasets obtained from same-sex samples were generated within a single batch (from nuclei extraction to barcoding, and sequencing), thereby reducing any batch effects. Still, the possibility of biological batch effects due to age differences or dataset-specific sources of heterogeneity, as well as inter-individual variation, could not be excluded. As requested by the reviewer, in order to improve the resolution of cell type identification and assess inter-animal variation, we have merged the snRNAseq datasets from same-sex samples using Seurat 3, and similarly the snATACseq datasets. The reason for merging rather than integrating the data was because integration methods like Harmony are designed to remove cross-sample variation and identify cell types across samples, but not to distinguish/evaluate batch effects or sample-to-sample variation. Once the merged data were labeled by individual animal, we could discern physiologically interpretable differences among the male as well as the female pituitaries. In the snRNAseq data, we distinguished a gonadotrope and a somatotrope cluster originating from one male animal. Both clusters were characterized by increased expression of immediate early genes, which may have resulted from an activating stimulus prior to death. In females, one gonadotrope and one lactotrope cluster observed in both

snRNAseq and snATACseq datasets stemmed from a single animal, reproducing the differences observed in other analyses (PLIER, SCENIC), and suggesting a different reproductive state in that female (#3). Fig. 2, Suppl. Fig.2, and Fig. 4 now show the merged samples and clusters by animal, as requested by the reviewer. Additionally, we provide UMI counts, mitochondrial gene expression, and ribosomal protein gene expression for the merged male and female samples (Suppl. Fig. 3), as well as for the individual samples (Suppl. Figs. 32-34). Our study highlights the benefit of using each individual pituitary as a separate dataset and combining those datasets to resolve whether distinct cell clusters in RNAseq and ATACseq data originate from different animals, whereas studies involving pooled pituitary samples cannot achieve such resolution.

2. For Fig 2c, there is an obvious effect of different UMI counts on cell clustering. This should be regressed out before PCA analysis.

As now indicated in the Results, when merging same-sex samples, there does not appear to be an effect of the UMI count on cell clustering:

“The distribution of UMI counts (Supplementary Fig. 3a,d), mitochondrial gene expression (Supplementary Fig. 3b,e), and ribosomal gene content (Supplementary Fig. 3c,f) in merged male and in merged female datasets showed some variation in some clusters, such as stem cells, yet this did not influence the identification of cell type clusters.”

To fully address the reviewer’s comment, in our original analysis we found that regressing out the UMI counts before PCA analysis did not affect the clustering analysis and had minimal effect on the identification of differentially expressed markers. SnRNAseq data is by nature sparse and affected by drop-outs. This sparsity affects the results of regression in a way that can be chaotic and unpredictable from gene to gene, stretching out the scale of certain genes while compressing that of others. The Seurat 3 clustering vignette no longer includes UMI count regression as a recommended step. As the two methods (merging of samples using Seurat vs. regression on individual samples) gave us identical clustering, given all considerations and current best practices, we chose to present the simpler Seurat analysis without the use of regression.

3. In line 140, the authors stated “Som 0 was unlikely to include doublets because the absolute number of transcripts (UMI counts) was on par with that of the other somatotropes”. It is unclear how the authors identify multiplets in snRNA-seq data (no detailed description in the method section). Because of separate clustering from individual animals, it is impossible to evaluate the overall relationship between clusters “Som 0” and “Db” from different animals. In Figure 2a, unlike mouse 1, where “Som 0” and “Db” clusters are well separated, they are quite close in tSNE plots of mouse 2 and 3. Also, it is necessary to show the UMI counts and mitochondrial percentage of these 2 clusters in mouse 2 and 3.

We acknowledge the reviewer's concern and now refer to the Supplementary Information in the Methods section for the step-by-step approach used to identify multiplets in the snRNAseq data analysis. This detailed approach is described in the Supplementary Methods, and Supplementary Figures have been added about the process.

Briefly, we identified cross-type doublets in two ways. First, doublets of the most abundant cell types form separate clusters, which are identified by our clustering analysis. This was the case for somatotrope-lactotrope doublets, for melanotrope doublets, and for stem cell doublets (with other cell types). We also observe that other doublets tend to separate out from their main cluster in t-SNE projection, even though we have not made use of the projection to identify them. While same type doublets are much more challenging to identify, they have much less impact on the results as they do not pollute cell-type expression with a different set of genes. Therefore, we have not made an attempt to exclude them other than with a liberal upper limit UMI cut of 45,000.

As requested by the reviewer, we now show in Supplementary Figs. 32-34 the plots for UMI count, percentage of mitochondrial gene expression, and percentage of ribosomal protein gene expression for each individual pituitary sample.

4. The author detected 3 sub-types of somatotropes (Som0, 1, and 2) clusters in snRNA-seq datasets. “Som 0” in Figure 2 appeared to be labeled as “Somato-lactorope” in Supplementary Figure 2, which is consistent with the observation that “Som 0” co-expresses Gh/Sgcz/Prl. The proportion of these sub-types should be summarized as in supplementary Table 2, and the differences in subtypes of somatotypes should be systematically examined and discussed based on the experimental methods (e.g. 3 subtypes of snRNA-seq in Figure 2a, 2 subtypes of snATAC-seq in Figure 3a and 1 cluster of sn methylation data in Figure 3b).

The reviewer raises a sensible point. The cluster Som0 indeed corresponds most likely to lactosomatotropes, which has been clarified in the Results as follows:

“While the pattern of expression in these cells was distinct from the rare lactotrope/somatotrope doublets, their expression of lactotrope, somatotrope and lactosomatrope genes, such as *Sgcz* and *Nrxn3*, suggests that this cluster may be comprised of lactosomatotropes (Supplementary Fig. 4). These putative lactosomatotropes were not as distinctly separated in the snATACseq datasets.”

The proportions of the somatotrope subtypes (Som0, Som1, and Som2) were determined for each individual pituitary sample in each dataset (a, snRNAseq; b, snATACseq), and are now presented in Supplementary Table 3 for the merged by sex analysis, Supplementary Table 4,5 for each individual sample, and in Supplementary Table 5 for the integrated analysis of the individual samples. No somatotrope subtype was identified in the sn methylation dataset.

In the merged data analysis (Fig. 2 for male pituitaries and Suppl. Fig.2 for female pituitaries), there was indeed variation in the number of somatotrope subclusters identified by snRNAseq vs. snATACseq vs. sn methylation. Additionally, by examining the differing contribution of data from individual pituitaries to the cell type clustering, we could discriminate subclusters arising from a single animal, e.g. a somatotrope subcluster from male #3 in the RNAseq data, and a somatotrope subcluster from female #3 in the ATACseq data. Altogether, we detected more cell clusters by snRNAseq (18 in male, 18 in female) than by snATACseq (11 in male, 15 in female), and the fewest by sn methylation analysis (9 in male). We propose in the Discussion that this variation reflects a continuum between transient cell states and developmentally determined cell types. While transient cell states manifest by changes in gene expression programs, cell types represent relatively stable developmentally encoded changes in expression, chromatin structure and methylation.

5. For all the reported cell types in Figure 4f, what is the relative percentage of these cell-types in pituitary based on integrated multi-omics datasets? Can the authors identify molecular differences for somatotrope subtypes (0, 1 and 2) in snATAC-seq and snmC-seq2 datasets?

The reviewer has a valid point. We have added Supplementary Table 6 based on the integrated analysis of the individual samples which provides the relative percentages of all cell types and somatotrope subtypes based on the integrated multi-omics data analysis of the individual samples. A similar table has been added for the merged analysis (Fig.4, Supplementary Table 3). The individual samples are presented in Supplementary Table 4.

We did identify molecular differences between the somatotrope subtypes both in the snRNAseq and in the snATACseq datasets. These differences are illustrated in Supplementary Figures 4 and 5 and reported in the Results as follows:

“The somatotrope clusters (Som) in males and females showed polarity in both snRNAseq and snATACseq, although the patterns leading to the gradients involved different genes. In the snRNAseq data, *Lhfpl3* and *Dpp10* were increased at one somatotrope cluster pole and *Tcerg11* at the other (Supplementary Fig. 4a-d). However, the genes whose promoter accessibility differentiated the two poles in the snATACseq included *Rem1* for one pole, and *Ascl2* and *Oit1* for the other (Supplementary Fig. 5). The cluster labeled as Som0 in snRNAseq data from males and females was located between the lactotrope and the somatotrope clusters on the UMAP projection (Fig. 4, Supplementary Fig.4). While the pattern of expression in these cells was distinct from the rare lactotrope/somatotrope doublets (Supplementary Fig. 4a), their expression of lactotrope, somatotrope and lactosomatrope genes, such as *Sgcz* and *Nrxn3*, suggests that this cluster may be comprised of lactosomatotropes (Supplementary Fig. 4a,e). These putative lactosomatotropes were not as distinctly separated in the snATACseq datasets. In females, we also observed a distinct cluster of lactotropes and somatotropes that showed chromatin accessibility for proliferating cell markers (Supplementary Fig. 2b) not seen in males (Fig. 2b). Accordingly, we labeled these two clusters as representing proliferating lactotropes and proliferating somatotropes (Supplementary Table 3).”

6. In Figure 3, why only male mice were analyzed for single-cell DNA methylation? Which genomic regions (promoter or gene body) are used for calculating the methylation level for marker genes in Figure 3c? Are these CpG or non-CpG methylation? These information are missing in the figure legend and method sections.

The reason why only male mice were analyzed by sn methylation is because the estrous cycle in female mice introduces additional heterogeneity between animals. As the current methodology used for sn methylation requires sample pooling to reach a sufficient number of nuclei, we had to pool a minimum of 30 individual pituitaries. Pooling 30 heterogeneous females into a single pool would have been an issue for data interpretation, therefore we chose to only analyze male pituitaries. We have added the following sentence in the Results:

“While the snRNAseq and snATACseq assays can be performed on single mouse pituitaries, the sn methylation method requires pooling pituitaries from many animals. Because of this limitation, we performed this assay only in males in view of the potentially increased complexity arising from estrous cycle stage differences among females.”

As indicated in the Results, for cell type identification, we determined the CpG methylation levels of pituitary marker genes at the promoter region. Non-CpG methylation was rare in our mouse pituitary samples, and thus disregarded. The following sentence has been added in the Results:

“In contrast with brain^{26, 27}, non-CG DNA methylation (mCH) was rare in pituitary, and was not used for cell type assignment.”

Similar information has been added in the Figure legends and in the Methods, as suggested by the reviewer.

7. In Figure 7b-c, many major cell-types (e.g. Somatotropes, Lactotropes) are separated into multiple sub-clusters, which seem to be mainly driven by batch effects rather than gender. For instance, it seems Lac from female #3 is separated away from other two females (#1/2) and is closes to male Lac. How did the authors integrate snRNA-seq datasets from different donors in Figure 7c?

It is crucial to emphasize that in each assay (i.e. snRNAseq or snATACseq), same-sex samples were processed simultaneously from nuclei extraction through sequencing, thus excluding any batch effect among same-sex samples. This has been clarified (twice) in the Methods:

“Same-sex samples were processed together from nuclei isolation to sequencing, with the three female samples processed one day, and the three male samples processed on another day.”

To improve the resolution of cell type identification in each omic modality, we have conducted an analysis of merged same-sex datasets (see Fig. 2 for males and Suppl. Fig. 2 for females).

Furthermore, labeling the merged data by individual pituitary has enabled us to evaluate the differing contribution from single animals to the cell type clustering (see Fig. 4). In this way, we determined that the snRNAseq dataset (as well as the snATACseq dataset) from female #3 contributed to an additional gonadotrope and lactotrope cluster. Notably, the estrogen-responsive genes *Greb1* and *Crhbp* were decreased in that lactotrope cluster relative to the lactotrope clusters associated with the other two females (Supplementary Fig. 16). Consistent with this finding, our LV analysis also revealed that female #3 differed from the other two females in that it showed a male-like pattern, including low expression levels of two estrogen-regulated genes (Fig. 7b and Supplementary Fig. 21a). As females were randomly selected with no determination of cycle status, our data suggest the identification in female #3 of a stage where estradiol is low, e.g. an absence of cycling. Our old Fig. 7c was in line with these observations and supported the male-like pattern of female #3. However, we now refer the reviewer to the merged analysis of male and female datasets, as it unambiguously brings to light animal-to-animal differences.

8. In figure 9, to support the causal relationship between the point mutations (rs11031006) and Fshb expression, the following experiments should be performed: a) The co-access between the peak/locus and Fshb promoter should be re-examined in the in vitro cell lines first; b) The point mutation (G to A/C/T) should be generated to examine the effect of Fshb expression rather than the deletion of “TATTT” in front of rs11031006.

Although the reviewer has a legitimate point, it is beyond the scope of the current paper to generate a CRISPR point mutation line and analyze the co-accessibility between the peak/locus and the *Fshb* promoter in cell lines. The analysis of the FSHB locus is mainly intended to illustrate the usefulness of our atlas in providing a jumping off point for additional study. It is not an in-depth study of this locus, which indeed would detract from the overall value of this publication. Nonetheless, we have included additional data from two additional CRISPR clones to address concerns about CRISPR off-target effects. The identification of similar FSH enhancing responses to different CRISPR modifications, along with results reported from human genetics, in aggregate strongly support the likely importance of this locus in modulating FSH expression. It should be noted that the rs11031006 SNP is in tight linkage disequilibrium with two other variants that span a large region⁹. However, rs11031006 was the most significant SNP in GWAS of polycystic ovary syndrome, and the region around rs11031006 is conserved across species. Therefore, it was chosen for further studies.

We have revised the Results to include the data from those additional CRISPR experiments, which are depicted in Supplementary Fig. 19. Finally, we indicate that following the posting of our study preprint², another study recently validated our hypothesis¹, which reinforces the value of our atlas for identifying putative regulatory regions that have a clinical relevance. The text now reads:

“Two other CRISPR modifications at this site (i. a T deletion/ TATT insertion; ii. a TTT deletion) each led to significant increases in *Fshb* expression following GnRH/activin A treatment, albeit to a lower extent than that observed in the TTATTT-deleted gonadotrope cell line (Supplementary Fig. 19). These findings suggest that the reproductive phenotype associated with human SNP rs11031006 results from altered regulatory control of the FSHB gene via this co-accessible upstream domain, and support the value of co-accessibility analysis, available on the atlas web portal, for the identification of putative cis-regulatory domains for any genes of interest. Following the posting of a preprint of our study², another group independently validated our hypothesis, providing further support for the use of our atlas to identify clinically important new putative regulatory domains¹.”

9. In general, the figures look very preliminary and lack essential information. For instance, in Figure 5, the scale (y-axis) and genomic coordinates are missing for all genomic tracks. In Figure 6, the labels are not explained. What is color-scale of heatmaps? What is the meaning of B-score? The gene names next to the heatmap are too small.

We have made considerable figure changes:

1. Fig. 2 now depicts the merged analysis of the male datasets.
2. Fig. 3 (previously Figure 5) has been improved. The genomic coordinates have been re-inserted for clarity, and the y-axis scales are now indicated at the top right corner for each pituitary marker.

We have edited the legends to address the reviewer’s comments.

3. Fig. 4 illustrates the merged analysis of same-sex samples with an overlay of the individual samples, which are color-coded.
4. Fig. 5 now includes a representative display of our web portal output.
5. Fig. 7 (previously Figure 6) has been improved. It now comprises a schematic/overview of LV identification by PLIER analysis. The color scale of heatmaps represents differential expression of a given gene across all cell types in all 6 samples (i.e. a row). Each row is scaled (subtract the mean of the row and divide by the standard deviation). Each heatmap has a scale bar, as defined in the figure legend.

In Fig. 7 (old fig.6), the labels for the B score plots were already provided.

6. Gene names and B score plots are now displayed in Suppl. Fig. 21. The names of the LVs in the main figure are now big enough to read. A paragraph defining the B-score is now provided in the Supplementary Methods, under PLIER analysis:

“B is a PLIER-derived expression value for the genes associated with a given LV across the different samples. It can be treated similar to average expression, weighted by gene association with the LV. Technically, B is a matrix of size #LVs x #Samples. It is one of two matrices in PLIER, along with Z of size # of genes x #LVs. The goal of PLIER is to find values of B and Z that minimize the equation $\|Y - Z*B\|$ where Y is our data matrix of size #genes x #samples. So PLIER finds a suitable number of LVs that can be used to connect the genes and samples and

accurately estimate our data matrix. We already cited this reference in our main text, but PLIER variables are described in detail in ¹⁰.”

7. Fig. 8 has been revised to improve clarity. An improved schematic of the model of cell type-specific regulon control mechanism is now presented in Fig. 9.

Minor points:

1. The Fig. 2c-d are identical to Supplementary Figure 4 b-c.

We thank the reviewer for pointing out this unintended redundancy. We have removed panels a-c from Supplementary Figure 4 and edited the legend accordingly. The legend now reads:

“Analysis of the somatotrope gradient cluster and the link between somatotropes and lactotropes via somatotrope 0 (som 0) in a single male sample. a. Heatmap of the lactotropes, somatotrope sub-types, and doublets showing the differences in gene expression. b, c. Feature plots showing *Tcerg11* expression at the Som 1 pole of the somatotrope cluster (b), and *Lhfp13* expression at the Som 2 pole (c). Refer to Supplementary Figures 32-34 for the tSNE, UMI, and mitochondrial gene level analysis for this same sample.”

2. In line 155-156, Supplementary Figure 8 appears in front of Supplementary Figure 7.

We thank the reviewer for pointing out this mistake. The figures and supplementary figures have been changed and many were added. We made sure to have them in the right order.

3. In line 157, Supplementary figure 8b does not support the claim “a canonical fragment-size distribution with clearly resolved mono- and multi-nucleosomal modes”.

As pointed out by the reviewer, we do not provide the snATACseq fragment size distribution in Supplementary Table 2b (formerly Supplementary Figure 8b). Therefore, the claim has been deleted from the Results. The text now reads:

“Whether the ATACseq data were derived from nuclei from snap-frozen pituitaries (Supplementary Fig. 13) or from cells from cryopreserved dissociated pituitaries (Supplementary Fig. 14), they exhibited a high signal-to-noise ratio at transcription start sites (TSS; see promoter of genes in Supplementary Fig. 14).”

4. In line 161, it should be “30,000 fragments per nuclues” not “30,000 peaks per nucleus”.

We thank the reviewer for mentioning this typo. We have edited the sentence as follows:

“An average of 5,000 nuclei were analyzed per pituitary sample, yielding approximately 30,000 fragments per nucleus (Supplementary Table 2b).”

5. For Supplementary Figure 12, it is necessary to show the scatterplot between methylation and expression of 331 FOXL2 regulon genes as in Figure 7e. Based on the association between gene expression and accessibility as well as between gene expression and TSS methylation, it does not support the strong conclusion that “These results indicate that promoter accessibility is a fundamental determinant of activity of FOXL2 regulon genes within the gonadotrope, whereas methylation has a more limited influence” in line 315-317.

We now provide the additional scatter plot requested by the reviewer in Supplementary Fig. 26a (formerly Supplementary Fig. 12). This scatter plot shows the relative methylation level in the promoter region (500 bp upstream of TSS) and the relative RNA expression of the Foxl2 targets in gonadotropes.

We have edited our concluding sentence but stand with the conclusion that:

“These results indicate that sex specific regulons encompass different biological functions in males and females and that promoter accessibility is a fundamental determinant of the sex differences in regulon activity.”

6. Lines 575 and 586, cited wrong reference. Ref#39 is not related to clustering of sn Methylation data sets.

There was indeed an error in the reference provided for the sn methylation dataset in the Methods section: reference #38 was mistakenly inserted instead of reference #57 (now #68. Luo C, et al. Single nucleus multi-omics links human cortical cell regulatory genome diversity to disease risk variants. Biorxiv <https://doi.org/10.1101/2019.12.11.873398>, (2019)). This has been corrected in our reference manager.

7. For “Mac” clusters in Supplementary Figure 6b, it is better to show some well-known markers of macrophages in the heatmap.

A heatmap is provided in Supplementary Fig. 12 that shows the markers that were differentially expressed in the macrophage cluster. The heatmap may not be showing the most well-known macrophage markers, as it displays the top markers that differentially separate the clusters. Nevertheless, top markers in the macrophage cluster included genes with high macrophage specificity, such as: *ApoE*, *Clqa*, *Clqb*, *Clqc*, *Ms4a7*, *Adgre1*. These genes, which had an average logFC ~ 2 - 3, were used to identify the cluster as macrophages.

8. In Supplementary Table 1 and Figure S1, the authors described sc/snRNA-seq and scATAC-seq experiments with different sample processing methods (cells vs. Nuclei), tissue dissociation protocols (gentle vs standard) and 10x chemistries (v2 vs. v3). For more meaningful comparisons, the authors should provide more information (#read, #UMI, #gene detected per cell/nucleus, etc.) in Table S1. Also, essential information and quality controls of dissociation protocols should be included in the method section. For example, In the Supplementary Information, the section of “Standard protocol used for dissociation”, what is the concentration of collagenase used.

1. As suggested by the reviewer, we now provide in Supplementary Table 1 all the quality control metrics of the samples processed during the optimization phase. This allows the reader to compare the different methods tested prior to adopting the snap-freeze and nuclei extraction approach. The updated Supplementary Table 1 includes 3 panels: a. list of all the samples analyzed in the study; b. sc/snRNAseq metrics for samples that were processed during the optimization phase; c. scATACseq metrics for samples that were processed during the optimization phase.

2. The following sub-section has been added in the Supplementary Methods to describe the steps taken to assess sample quality at the different stages of the protocol and during optimization of the protocol:

“Quality control (QC) assessment

We tested different protocols to assess the best approach for building a murine single-cell pituitary atlas. We assessed the quality of our samples and data by measuring cell viability for cryopreserved dissociated pituitary cells, and by observing the nuclei shapes and sizes during nuclei isolation and counts. We followed standard QC recommended in the 10x protocols, making sure the emulsion step was not compromised, performing bionalyzer assessment of the samples after the cDNA amplification stage for sc/sn RNAseq, as well as after library preparation for all samples (sc/sn RNAseq and sc ATACseq). After sequencing standard metrics were assessed in Cell Ranger to make sure our samples were passing QC. Sequencing metrics are presented for all samples in supplementary tables and figures.”

3. Supplementary Figs. 8 and 9 illustrate the scRNAseq vs. snRNAseq analysis of the same gently dissociated and cryopreserved female pituitary. In the snRNAseq dataset, we do not see the low UMI clusters that are observed in the scRNAseq dataset. These low UMI clusters are likely indicative of damaged/dying cells (based on their high mitochondrial content). We also noticed that when using cryopreserved cells, immediate early genes were activated in some clusters, and we had no way to determine whether this was an artefact associated with dissociating/freezing/thawing the cells, or a true biological feature. Using snap-frozen pituitaries

eliminates the dissociation aspect and potential mechanical stress on the cells. Using nuclei instead of whole cells excludes the presence of damaged cells in the analysis.

4. In the Supplementary Methods, we now provide the concentrations of collagenase and pancreatin that were used in the *Standard protocol used for dissociation*. The text now reads: “Upon collection, single pituitaries were digested with 40 ml of 1.5mg/ml collagenase (Sigma, Type I-C0130) in Low Ca²⁺HBSS (pH 7.2-7.4) and dispersed with 4ml of 2.5mg/ml pancreatin (10X; Sigma #P3292) in 0.85% NaCl with 36 µl HBSS (NO CALCIUM)”.

Reviewer #3 (Remarks to the Author):

In this manuscript, Ruf-Zamojski and colleagues present the first large-scale single-nucleus multi-omics atlas of pituitary cells of adult mice. The scope of this work is impressive, including single-nucleus RNAseq, ATACseq, and DNA methylation data from a total of more than 70,000 pituitary cells from both male and female adult mice. By integrating these data, the authors are able to pinpoint differential regulatory gene networks/regulons of genes expressed in multiple pituitary cell-types, such as the transcription factor Foxp2, as well as functionally test a gonadotrope-accessible SNP associated with human fertility. Finally, the data have been made openly accessible through an interactive web portal, and these data will be of great interest and utility to the research community.

In identifying gonadotrope regulons of Foxl2, is there any functional data to support their role in gonadotrope function or otherwise address the role of these genes and pathways? It is difficult to interpret the biological interpretation of the regulons given the large number of genes and wide variety of biological terms/pathways.

The reviewer makes a valid point, which is to ask whether we can provide functional data supporting the role of the Foxl2 regulon in the gonadotropes. Firstly, we want to emphasize that we focused on the Foxl2 regulon because its expression is gonadotrope-specific (see Fig. 8b). Secondly, our data suggest that Foxl2 differentially regulates chromatin accessibility of its target genes in gonadotropes compared with other cell types. Thirdly, previous conditional knockout of Foxl2 in murine gonadotropes demonstrated the critical role of Foxl2 in FSH synthesis in vivo, and its subsequent impact on fertility¹¹. Activin-stimulated *Fshb* transcription was also shown to be impaired in immortalized gonadotrope cells^{12, 13} and in primary pituitary cultures¹¹. While further investigation of these Foxl2 knockout mice at the single-cell level may help to elucidate the gonadotrope-specific molecular mechanisms underlying *Fshb* gene regulation by Foxl2, it is beyond the scope of the current study. The text now reads:

“FOXL2 is essential in basal and activin-stimulated *Fshb* transcription and FSH β expression in the gonadotropes, and subsequently impacts mouse fertility^{50, 51, 52, 53, 54}”

*Similarly, can you provide any functional data to suggest how *Foxp2* differentially regulates different genes in different cell-types, e.g. by showing their differential chromatin accessibility or DNA methylation? This data would be strengthened if there were data to functionally test *Foxp2* expression in different pituitary cell lines to demonstrate context-dependent activity.*

Yes, our data suggest that FOXP2 differentially regulates chromatin accessibility of its target genes in gonadotropes vs. melanotropes vs. stem cells. Moreover, the finding that *Foxp2* is predominantly expressed in gonadotropes (Fig. 8b), and the recent identification of *Foxp2* as a novel gonadotrope-enriched marker both by scRNAseq and by immunohistochemistry in vivo¹⁴ support its functional role in the gonadotrope cell type. While previous studies reported the importance of *Foxp2* in the brain and its association with speech and language disorder¹⁵, no studies have looked at the role of *Foxp2* in the pituitary. Further investigation in *Foxp2*-null mice may provide insights into its potential role in the pituitary gland¹⁶.

*In Fig 7e, although there is an overall positive correlation between chromatin accessibility and gene expression, there are some highlighted gonadotrope genes that are high accessibility-low expression (*Cga/Fshb*) and vice-versa (*Nr5a1*). What do you make of these outliers? In Fig 7f, what is the definition of ‘relatively low’ and ‘relatively high’ expression? In identifying *Foxl2* regulon genes in gonadotropes (Fig 7g/h), could you describe whether the expression of these genes is correlated with high or low expression of *Foxl2* itself?*

The reviewer indeed raises pertinent questions about the *Foxl2* regulon, which is now illustrated in Suppl. Fig. 22c-f (formerly Fig. 7e-h). *Cga* is expressed at a high level in gonadotropes. However, because *Cga* is expressed in every cell type at background level, its relative expression in gonadotropes is not as high as *Fshb*, which is gonadotrope-specific. Moreover, the reason why relative *Fshb* expression is modest is because it is expressed in only a few cells (i.e. a few gonadotropes). *Nr5a1* gene shows high expression despite low/medium promoter accessibility. One possible explanation is that the *Nr5a1* promoter may be occupied by large regulatory complexes blocking access to transposase¹⁷. Alternatively, its promoter could show higher nucleosome occupancy¹⁸.

With respect to Suppl. Fig. 22e/f (formerly Fig. 7g/h), a detailed explanation for how we identify target genes that are “on” in each regulon and each cell type is provided in the Methods. Briefly, we set a threshold for defining relatively high and relatively low expression, and in the analysis combining multiple samples (e.g. all males), a gene is identified as “high” if it is “high” in at least one sample. The text now reads as:

“To study the target genes specifically turned on in each regulon and each cell type, we identified a target gene as specifically highly expressed in a cell type if it satisfied either of these two conditions: (1) its relative averaged normalized RNA expression in this cell type vs. the whole dataset was greater than 2, and it had non-zero UMI counts in at least 10% of the cells in this cell type; and (2) it had non-zero UMI counts in at least 50% of the cells, and its relative averaged normalized RNA expression in this cell type was greater than 1. The rationale behind this was to avoid genes being classified as “high” if they were universally lowly expressed across all cells but showed high relative expression because of noise. In the analysis involving multiple samples such as all males, a target gene was defined as specifically highly expressed in a cell type if it was specifically highly expressed in the cell type in at least one sample.”

Lastly, the target genes in the FOXL2 regulon were correlated with the expression of *Foxl2* itself. In fact, in the first step of the SCENIC pipeline, a gene has to show correlation with *Foxl2* to be identified as a potential target. This relationship is also shown in the histogram below, where all the correlations between *Foxl2* and its target genes were greater than 0.

In Fig 9d, why do the CRISPR deletions not cover the SNP location? Do humans bearing variation at this SNP show increased basal and/or secreted Fshb levels? Discuss how increased Fshb might relate to PCOS and endometriosis phenotypes in humans.

Fig. 9d is now Fig. 6d. It should be noted that the rs11031006 SNP is in tight linkage disequilibrium with two additional variants that span a large region⁹. Based on genome-wide association studies of polycystic ovary syndrome (PCOS), rs11031006, which lies near the FSHB gene, is the most strongly associated with PCOS⁹, and the region around rs11031006 is conserved across species (see Fig. 6b,c). Additionally, the rs11031006 variant is in the proximity of H3K4Me1 sites in several human cell lines, including HUES48 and iPS-15b cells. It is also located approximately 500 bases downstream of H3K4Me1 sites and 1,500 to 2,000 bases upstream of both H3K4Me1 and H3K27Ac sites and of open chromatin (identified by DNase-Seq). Both the H3K4Me1 and H3K27Ac histone marks are associated with distal enhancers, suggesting a regulatory role for the region around the rs11031006 SNP. Our CRISPR deletions were designed to disrupt the conserved region in the vicinity of rs11031006 in the mouse genome, and determine whether this region directly regulates the nearest gene *Fshb*. Our preliminary experiments demonstrated that this region regulates *Fshb* gene expression. Additionally, following the posting of a preprint of our study², another group independently reported that the rs11031006 minor allele upregulates FSHB transcription via increased SF1 binding to the *Fshb* distal enhancer¹.

The PCOS-susceptibility variant rs11031006-A is strongly associated with lower FSH levels, higher LH levels, and higher LH/FSH ratio¹⁹. Disruption of the region in the current study resulted in increased FSH expression and secretion, consistent with a regulatory, possibly repressive role of this region in FSHB expression. The association with higher LH concentrations likely results from greater availability of the alpha subunit that binds to both FSH β and LH β subunits, or less estradiol feedback.

An elevated LH:FSH ratio is often observed in women with PCOS. Higher LH concentrations result in increased androgen stimulation from the theca cells. The lower FSH is also consistent with the large number of small follicles (called cysts) arrested in development, which characterize the polycystic ovary. These follicles grow readily upon stimulation with FSH in fertility treatment, demonstrating the relative lack of FSH in these women. In contrast to its association with PCOS, the rs11031006-A allele is associated with lower risk of endometriosis²⁰. As FSH stimulates aromatase and estradiol production, a relatively lower FSH could result in lower estradiol production, less endometrial lining, and less endometrial tissue to deposit into the pelvis.

The amount of data presented can cause the data to be difficult to see in places. Figures 2 and 3 describe the snRNAseq and snATACseq data individually, although they are not subsequently analyzed as separate datasets. In both, why do the expression/accessibility plots display data from mouse 1 only instead of all samples combined? In Fig 3c, the color scale of dark blue to yellow can be difficult to see, particularly for yellow on white. In Fig 5, the peaks under the + and - signs on the DNA methylation data are very difficult to see and there are several that I am unable to discern a difference from other cell-types, e.g. the hypomethylated

regions in Cga in lactotropes (Fig 5a). The key for Fig 5 may be simpler if ordered top-to-bottom in the same order they appear in the tracks rather than a 2x4 arrangement.

1. We have addressed the reviewer's critique about Figs. 2 and 3, as we now present a merged analysis of same-sex datasets in Fig. 2, Suppl. Fig. 2, and Fig. 4. A detailed explanation is provided below. Individual tSNE plots have been moved to the Supplementary Information (see Suppl. Figs. 10 and 13).

We recognize the importance of integrating snRNAseq data across different conditions, technologies, or species in order to eliminate batch effects^{6,7,8}. We need to emphasize, however, that the snRNAseq and snATACseq datasets obtained from same-sex samples were generated within a single batch (from nuclei extraction to barcoding, and sequencing), thereby reducing any batch effects. Still, the possibility of biological batch effects due to age differences or dataset-specific sources of heterogeneity, as well as inter-individual variation, could not be excluded. Therefore, in order to improve the resolution of cell type identification and assess inter-animal variation, we have merged the snRNAseq datasets from same-sex samples using Seurat 3, and similarly the snATACseq datasets (see Fig. 2 for males and Suppl. Fig. 2 for females). Additionally, we provide UMI counts, mitochondrial gene expression, and ribosomal protein gene expression for the merged male and female samples (Suppl. Fig. 3), as well as for the individual samples (Suppl. Figs. 32-34). Labeling the merged data by individual animal (Fig. 4) enabled us to detect physiologically interpretable differences among the male as well as the female pituitaries. For instance, in the snRNAseq data we distinguished a gonadotrope and a somatotrope cluster originating from one male animal. Both clusters were characterized by increased expression of immediate early genes, which may have resulted from an activating stimulus prior to death. In females, one gonadotrope and one lactotrope cluster observed in both snRNAseq and snATACseq datasets stemmed from a single animal, reproducing the differences observed in other analyses (PLIER, SCENIC), and suggesting a different reproductive state in that female (#3).

2. As per the reviewer's request, we have significantly improved Fig. 3 (formerly Fig. 5).

We have increased the size and changed the color of the + and - signs. The signs have been placed above each methylation track to better guide the reader. The key has been modified to match the order of appearance of the tracks, and is now displayed as a single column.

In Fig 6, inference of female mouse 3 as being in non-cycling may be premature given the possibility of individual variety. Could female mouse #3 be in a different estrous cycle stage? Does the data from Boehm on gene expression at different stages of the estrous cycle informative in this regard?

We acknowledge that it is premature to claim that female #3 is non-cycling. Nevertheless, analysis of the merged female datasets supports the notion that female #3 may either be non-cycling or at a different stage of the estrous cycle, as recapitulated below and indicated in the Results.

Labeling the merged data by individual pituitary enabled us to evaluate the differing contribution from single animals to the cell type clustering (see Fig. 4). In this way, we determined that the snRNAseq dataset (as well as the snATACseq dataset) from female #3 contributed to an additional gonadotrope and lactotrope cluster. Remarkably, that gonadotrope cluster exhibited increased *Fshb* gene expression and promoter accessibility, increased *Pitx1* expression, and decreased expression of key reproductive genes including *Greb1*, *Gnaq*, and *Pgr* in comparison with the gonadotrope clusters associated with the other two females (Supplementary Fig. 16). In addition, the lactotrope cluster from female #3 showed decreased expression of the reproductive genes *Greb1* and *Crhbp* relative to the lactotrope clusters associated with the other two females (Supplementary Fig. 16). Thus, these findings were consistent with our LV analysis indicating that female #3 had a male-like pattern (Fig. 7b and Supplementary Fig. 21a). Altogether, our data suggest that female #3 is either non-cycling or at a different stage of the estrous cycle.

Qiao 2016²¹ analyzed the genes that were differentially expressed in diestrus, proestrus, lactating, and juvenile female mice. Although it would be interesting to compare our data with those differentially expressed genes, genetic and age differences between the females in our study and that of Boehm's group may make interpretation difficult, such that we would remain speculative about the estrous cycle stage. In future multi omics studies of single animals, we will determine the stage of the estrous cycle before sacrificing the mice.

Line 133: typo, SNN is shared nearest neighbor

Line 1065: typo, LV1tacet should be LV1atac

We thank the reviewer for pointing out these typos. We have now corrected them.

References:

1. Bohaczuk SC, Thackray VG, Shen J, Skowronska-Krawczyk D, Mellon PL. FSHB Transcription is Regulated by a Novel 5' Distal Enhancer with a Fertility-Associated Single Nucleotide Polymorphism. *Endocrinology*, (2020).
2. Ruf-Zamojski F, *et al.* Single nucleus multi-omics regulatory atlas of the murine pituitary. *bioRxiv*, 2020.2006.2006.138024 (2020).
3. Kim M, Costello J. DNA methylation: an epigenetic mark of cellular memory. *Exp Mol Med* **49**, e322 (2017).

4. Luo C, *et al.* Single-cell methylomes identify neuronal subtypes and regulatory elements in mammalian cortex. *Science* **357**, 600-604 (2017).
5. Ludwig CH, Bintu L. Mapping chromatin modifications at the single cell level. *Development* **146**, (2019).
6. Butler A, Hoffman P, Smibert P, Papalexi E, Satija R. Integrating single-cell transcriptomic data across different conditions, technologies, and species. *Nat Biotechnol* **36**, 411-420 (2018).
7. Haghverdi L, Lun ATL, Morgan MD, Marioni JC. Batch effects in single-cell RNA-sequencing data are corrected by matching mutual nearest neighbors. *Nat Biotechnol* **36**, 421-427 (2018).
8. Stuart T, Satija R. Integrative single-cell analysis. *Nat Rev Genet* **20**, 257-272 (2019).
9. Hayes MG, *et al.* Genome-wide association of polycystic ovary syndrome implicates alterations in gonadotropin secretion in European ancestry populations. *Nat Commun* **6**, 7502 (2015).
10. Mao W, Zaslavsky E, Hartmann BM, Sealfon SC, Chikina M. Pathway-level information extractor (PLIER) for gene expression data. *Nat Methods* **16**, 607-610 (2019).
11. Tran S, *et al.* Impaired fertility and FSH synthesis in gonadotrope-specific Foxl2 knockout mice. *Mol Endocrinol* **27**, 407-421 (2013).
12. Lamba P, Fortin J, Tran S, Wang Y, Bernard DJ. A novel role for the forkhead transcription factor FOXL2 in activin A-regulated follicle-stimulating hormone beta subunit transcription. *Mol Endocrinol* **23**, 1001-1013 (2009).
13. Tran S, Lamba P, Wang Y, Bernard DJ. SMADs and FOXL2 synergistically regulate murine FSHbeta transcription via a conserved proximal promoter element. *Mol Endocrinol* **25**, 1170-1183 (2011).
14. Cheung LYM, *et al.* Single-Cell RNA Sequencing Reveals Novel Markers of Male Pituitary Stem Cells and Hormone-Producing Cell Types. *Endocrinology* **159**, 3910-3924 (2018).
15. Lai CS, Gerrelli D, Monaco AP, Fisher SE, Copp AJ. FOXP2 expression during brain development coincides with adult sites of pathology in a severe speech and language disorder. *Brain* **126**, 2455-2462 (2003).
16. French CA, Groszer M, Preece C, Coupe AM, Rajewsky K, Fisher SE. Generation of mice with a conditional Foxp2 null allele. *Genesis* **45**, 440-446 (2007).

17. Daugherty AC, Yeo RW, Buenrostro JD, Greenleaf WJ, Kundaje A, Brunet A. Chromatin accessibility dynamics reveal novel functional enhancers in *C. elegans*. *Genome Res* **27**, 2096-2107 (2017).
18. Jiang C, Pugh BF. Nucleosome positioning and gene regulation: advances through genomics. *Nat Rev Genet* **10**, 161-172 (2009).
19. Day FR, *et al.* Causal mechanisms and balancing selection inferred from genetic associations with polycystic ovary syndrome. *Nat Commun* **6**, 8464 (2015).
20. Matalliotakis M, *et al.* The role of gene polymorphisms in endometriosis. *Mol Med Rep* **16**, 5881-5886 (2017).
21. Qiao S, *et al.* Molecular Plasticity of Male and Female Murine Gonadotropes Revealed by mRNA Sequencing. *Endocrinology* **157**, 1082-1093 (2016).

REVIEWERS' COMMENTS

Reviewer #1 (Remarks to the Author):

The authors have adequately addressed this reviewer's comments.

Reviewer #2 (Remarks to the Author):

The authors have largely addressed our concerns in the revised manuscript.

Reviewer #3 (Remarks to the Author):

The authors have provided an exhaustive response to the points made by each reviewer. I judge these to be appropriate.

Sally Camper